# PAXX and its paralogs synergistically direct DNA polymerase λ activity in DNA repair

Andrew Craxton[1], Deeksha Munnur[1,2], Rebekah Jukes-Jones[1], George Skalka[1], Claudia Langlais[1], Kelvin Cain[1] & Michal Malewicz [1]

PAXX is a recently identified component of the nonhomologous end joining (NHEJ) DNA repair pathway. The molecular mechanisms of PAXX action remain largely unclear. Here we characterise the interactomes of PAXX and its paralogs, XLF and XRCC4, to show that these factors share the ability to interact with DNA polymerase λ (Pol λ), stimulate its activity and are required for recruitment of Pol λ to laser-induced DNA damage sites. Stimulation of Pol λ activity by XRCC4 paralogs requires a direct interaction between the SP/8 kDa domain of Pol λ and their N-terminal head domains to facilitate recognition of the 5′ end of substrate gaps. Furthermore, PAXX and XLF collaborate with Pol λ to promote joining of incompatible DNA ends and are redundant in supporting Pol λ function in vivo. Our findings identify Pol λ as a novel downstream effector of PAXX function and show XRCC4 paralogs act in synergy to regulate polymerase activity in NHEJ.

[1] MRC Toxicology Unit, Hodgkin Building, Lancaster Road, Leicester LE1 9HN, UK. [2] Present address: Sir William Dunn School of Pathology, South Parks Road, University of Oxford, Oxford, UK. Correspondence and requests for materials should be addressed to M.M. (email: mzm23@mrc-tox.cam.ac.uk)

DNA double-strand breaks (DSBs) are one of the most cytotoxic types of DNA damage in mammalian cells. Pathological DSBs can occur endogenously as a consequence of oxidative DNA damage, abortive action of nuclear enzymes involved in DNA metabolism or due to exogenous DNA damaging agents including ionising radiation (IR). Interestingly DSBs are also required for normal development such as RAG-dependent breaks occurring during V(D)J recombination[1–6]. Unrepaired or misrepaired DSBs cause genomic instability resulting in cell death, senescence and predisposition to cancers. Mammalian cells engage two major pathways to resolve DSBs, homologous recombination (HR) and NHEJ. HR utilises an intact sister chromatid as template to guide repair, which limits HR to the S and G2/M phases of the cell cycle[2]. In contrast, NHEJ directly rejoins DSBs and crucially does not require extensive homology. NHEJ occurs during all phases of the cell cycle, including the G1 phase when cells are uniquely dependent on NHEJ. NHEJ occurs in a series of stages and requires co-ordinated involvement of a repertoire of key proteins[4–6]. Initially, the DSB is sensed by Ku70/80 heterodimers leading to recruitment of DNA-PKcs (known together as the DNA-PK holoenzyme complex), activation of its protein kinase activity and tethering of the DNA termini. DNA-bound Ku serves as a platform for recruitment of various proteins including some with enzymatic activities required to process complex damaged DNA ends. These include the nuclease Artemis, polynucleotide kinase-phosphatase (PNKP), Werner (WRN) helicase and the family X DNA polymerases λ (Pol λ) and μ (Pol μ)[6]. DNA-bound Ku also recruits two structurally related proteins, XRCC4- and XRCC4-like factor (XLF/Cernunnos/NHEJ1), which independently from Ku can form long filaments facilitating alignment of DNA ends prior to ligation, the final step of NHEJ mediated by DNA Ligase IV (Lig IV)[4–9]. NHEJ reactions proceed via a variety of so-called "subpathways", which utilise various subsets of NHEJ proteins and differ in the way distinct DNA ends are processed prior to ligation[6]. Recently, an additional XRCC4-like protein, PAXX (Paralog of XRCC4 and XLF; also known as XLS or c9orf142) was identified[10–12]. XRCC4 and its paralogs consist of highly conserved N-terminal globular head domains, a centrally located coiled-coil and a C-terminal region. PAXX is required for resistance to IR-induced DNA damage and interacts with DNA-PK holoenzyme via protein–protein interactions with DNA-bound Ku heterodimers[10–12]. Studies using PAXX and XLF-deficient mice showed that PAXX and XLF share redundant functions, as unlike single knockouts most PAXX/XLF double knockout mice exhibit embryonic lethality associated with major defects in growth, lymphogenesis and increased neuronal cell death[13,14]. Importantly, PAXX also stimulated ligation of noncohesive DNA ends in a XLF-dependent manner[12]. These findings led us to hypothesise that PAXX may play a specific role in processing of non-compatible DNA ends. Such processing requires various factors including DNA polymerases. NHEJ specifically employs Pol λ and Pol μ, whose structures are characterised by a common protein fold with similar secondary structure[15,16]. While loss of either Pol λ or Pol μ alone resulted in a mild increase in IR sensitivity, cells deficient in both DNA polymerases were highly radiosensitive, consistent with the notion that these two DNA polymerases together are essential for efficient NHEJ[17].

To gain broader insight into interactions mediated by PAXX and its paralogs and to further investigate a role for PAXX in processing of non-compatible DNA ends, we characterise the interactome of PAXX, XLF and XRCC4. Our studies identify Pol λ as an abundant PAXX-interacting protein, which also interacts with XLF and XRCC4. PAXX and XLF form a complex with Pol λ via DNA-bound Ku heterodimers. This interaction requires the C-terminal region of PAXX or XLF. In addition, each XRCC4 family protein stimulates Pol λ-dependent gap-filling activity in vitro via direct interaction between the head domains and the Ser-Pro/8 kDa region respectively. Cell extracts depleted in PAXX, XLF or XRCC4 exhibit reduced Pol λ-dependent gap filling activity. Significantly, recruitment and retention of Pol λ at DNA damage sites in cells is also strongly reduced in PAXX, XLF or XRCC4-deficient cells. Thus, PAXX, XLF and XRCC4 synergise in the efficient DSB recruitment, substrate recognition and stimulation of Pol λ enzymatic activity during NHEJ.

## Results

**Analysis of the interactomes of PAXX and its paralogs.** First we performed comparative proteomic analysis of PAXX, XLF and XRCC4 interactomes using DNA-PKcs as a reference bait[10] and utilising cell lines stably overexpressing N-terminal FLAG-PAXX, -XLF or -XRCC4 proteins. We found bait proteins to be present in both nuclear soluble and insoluble fractions and used a nuclease (benzonase) to facilitate the release from the insoluble compartment (Supplementary Fig. 1b[18]). Next, we isolated proteins associated with PAXX, XLF, XRCC4 and DNA-PKcs from nucleoplasmic and benzonase-treated (soluble chromatin) fractions by anti-FLAG immunoprecipitation followed by elution with FLAG peptide (Figs. 1, 2, Supplementary Fig. 1a, c–f). Immunoblotting showed that PAXX and XLF associated with other NHEJ factors (DNA-PK holoenzyme, Lig IV), and PAXX preferentially associated with these NHEJ proteins in the soluble chromatin fraction (Supplementary Fig. 1e, f). In contrast, XRCC4 preferentially associated with Lig IV (Supplementary Fig. 1e-f)[12]. These results were also confirmed by mass spectrometry by revealing a preferential association of PAXX with key

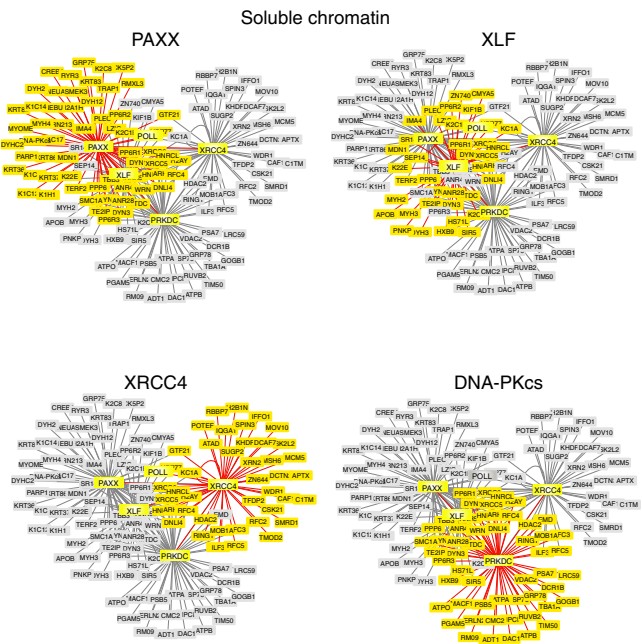

**Fig. 1** Comparative Analysis of the Proteomes of XRCC4 Family Proteins isolated from Soluble Chromatin. Interactome analysis using FLAG-tagged PAXX, -XLF, -XRCC4 and -DNA-PKcs as bait from the benzonase-treated soluble chromatin fraction of HEK293F cells was performed using Cytoscape. Proteins highlighted by shaded yellow boxes indicate proteins which interact with the indicated bait protein e.g. PAXX, XLF, XRCC4 or DNA-PKcs. Shaded grey boxes depict proteins identified in combined proteomes of XRCC4 family proteins and DNA-PKcs (PRKDC)

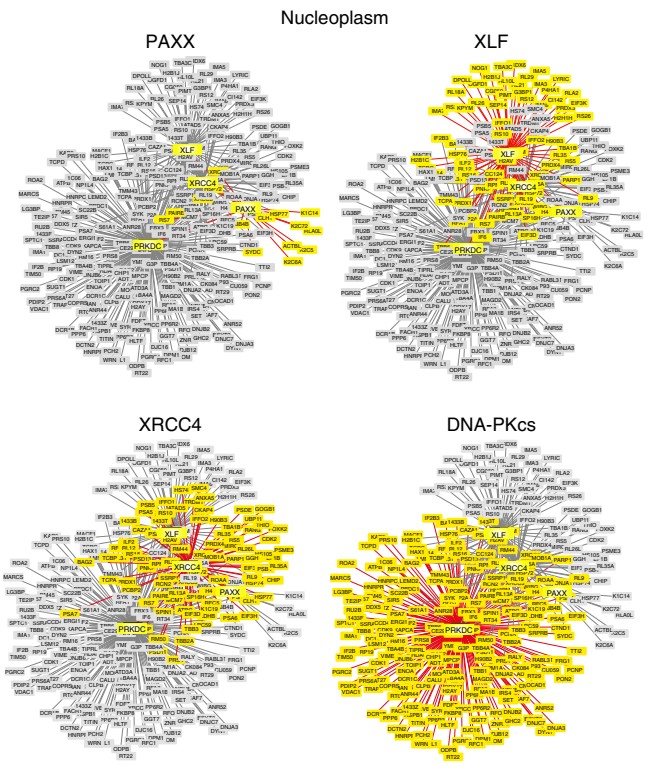

**Fig. 2** Comparative Analysis of the Nucleoplasmic Proteomes of XRCC4 Family Proteins. As described in Fig. 1, except the nucleoplasmic fraction was analysed

NHEJ factors in the soluble chromatin fraction (Supplementary Tables 6, 7 and Supplementary Data 1).

In addition to their interaction with core NHEJ proteins, PAXX or its paralogs also associated with a variety of NHEJ accessory factors including PNKP, APTX, WRN, PARP1 and Pol λ (Figs 1, 2, Supplementary Tables 6 and 7). Notably, PAXX bound Pol λ in the soluble chromatin fraction, which plays an important role during NHEJ to fill DNA gaps (Supplementary Tables 6, 7 and Supplementary Data 1)[4,5]. PAXX also associated with multiple subunits of the trimeric protein phosphatase 6 holoenzyme, which directly interacts with and regulate DNA-PK function (Fig. 1 and Supplementary Fig. 9)[19,20]. In addition, multiple dynamin members (DYN1, 2 and -3), TRF2/TERF2 and its interacting protein TERF2IP/RAP1, also co-purified with PAXX, XLF and DNA-PKcs but not XRCC4 (Fig. 1 and Supplementary Data 1). In contrast, relatively few proteins were shared between PAXX or XLF and XRCC4 in the soluble chromatin nuclear fraction (Fig. 1). We also identified a small cohort of proteins which appeared to selectively interact with PAXX or XLF (Supplementary Table 8). Consistent with the notion that DNA-PKcs is involved in regulating other cellular pathways independent from its role in DNA repair[21], DNA-PKcs interacted with a large cohort of additional proteins specific to this bait (Figs 1, 2, Supplementary Data 1).

**Interactions with DNA Polymerase λ.** As we identified Pol λ as an abundant PAXX-associated protein and PAXX has been reported to stimulate joining of noncomplementary DNA ends in a XLF-dependent manner in vitro[12,17], we investigated the interaction between PAXX and Pol λ. Pol λ co-purified with overexpressed FLAG-PAXX in HEK293 and U2OS cells (Supplementary Fig. 2a, b). Reciprocal immunoprecipitation with

FLAG-tagged Pol λ demonstrated interaction of PAXX and its paralogs (and other NHEJ factors) with Pol λ (Supplementary Fig. 2c, d). Furthermore, the Pol λ interactome included PAXX and its paralogs (Supplementary Table 9 and Supplementary Data 2). Importantly, immunoprecipitation of endogenous Pol λ showed that PAXX, XLF and XRCC4 co-purified with Pol λ in untreated cells or after ionising irradiation (Fig. 3a). Association of these endogenous proteins was further confirmed by reciprocal IPs of PAXX, XLF and XRCC4 with Pol λ (Fig. 3b). We also examined the effect of ethidium bromide (EtBr), which specifically disrupts interaction of proteins with DNA, on interaction of PAXX and its paralogs with Pol λ. EtBr blocked interaction of Pol λ with PAXX (Fig. 3c). Association of Pol λ with XLF or XRCC4 were less affected by EtBr, suggesting possible direct protein–protein contact (Fig. 3c).

The N-terminal BRCT domain of Pol λ and key conserved residues in α-helix 1 (Arg57, Leu60) of its BRCT domain are required for its DNA-dependent association with Ku/XRCC4/Lig IV in vitro[22–24]. To examine whether the Pol λ BRCT domain and its α-helix 1 were required for its interaction with PAXX in cells, we examined interaction of Pol λ-WT, -ΔBRCT and -R57A/L60A (RL) with DNA-bound Ku in vitro and also in cells (Fig. 3d, e, Supplementary Fig. 2e, f). Arg57/Leu60 within α-helix 1 of the Pol λ BRCT domain was required for its interaction with DNA-bound Ku in vitro (Fig. 3d)[22]. Moreover, association of Pol λ with PAXX and its paralogs, and Ku heterodimers required the BRCT domain-containing N-terminal region of Pol λ and its conserved BRCT α-helix 1 residues (Fig. 3e).

**Ku-dependence of DNA Polymerase λ interaction.** Our observation that the BRCT domain of Pol λ is required for its interaction with PAXX raised the possibility that DNA-bound Ku heterodimers facilitate interaction between PAXX and Pol λ. We explored this possibility using 30- or 90 bp dsDNA oligonucleotide Pol λ substrates containing 5nt gaps by EMSA[25]. PAXX exhibited no binding activity towards these DNA substrates (Fig. 4a, lanes 2 and 9; Supplementary Fig. 3a, lane 2). However, Pol λ formed a complex with a 5nt-gapped dsDNA substrate (Fig. 4a, lanes 3 and 10; Supplementary Fig. 3a, lane 3)[23,25]. Although PAXX did not supershift Pol λ-DNA complexes (Fig. 4a, lanes 4 and 11; Supplementary Fig. 3a, lanes 6–7), detectable association between PAXX and Pol λ was observed in GST pull downs, although reduced compared to XLF-Pol λ interaction (Supplementary Fig. 3b). Consistent with these results, a minor fraction of DNA-bound Pol λ was supershifted by XLF (Supplementary Fig. 3c). PAXX or Pol λ supershifted DNA-bound Ku70/80 complexes (Fig. 4a, lanes 5–7 and 12–14; Supplementary Fig. 3a, lanes 4, 5 and 8). Importantly, PAXX supershifted Pol λ-Ku70/80-DNA complexes with formation of a further retarded PAXX-Pol λ-Ku70/80-DNA complex (Fig. 4a, lanes 8 and 15; Supplementary Fig. 3a, lanes 9-10). To assess a role for PAXX-Ku interaction(s) in formation of PAXX-Pol λ-Ku70/80-DNA quaternary complexes, we generated a PAXX C-terminal Ku-binding mutant (PAXX-V199A/F201A (PAXX-VF)), Supplementary Fig. 3d[11]. PAXX-VF, in contrast to PAXX-WT, did not form a complex with DNA-bound Ku70/80 as shown (Fig. 4b)[11]. Consistent with a Ku-dependent interaction between PAXX and Pol λ, PAXX-WT but not PAXX-VF supershifted the Pol λ-Ku70/80-DNA complex (Fig. 4c).

As endogenous XLF also associated with Pol λ in cells (Fig. 3a, b), we also tested whether Ku mediates its interaction with Pol λ in vitro. Ku-dependent binding of XLF-WT to DNA was observed (Fig. 4d) as reported[26]. XLF-WT supershifted the Pol λ-Ku70/80-DNA complex but not the Pol λ-DNA complex, suggesting that Ku principally mediates association between XLF

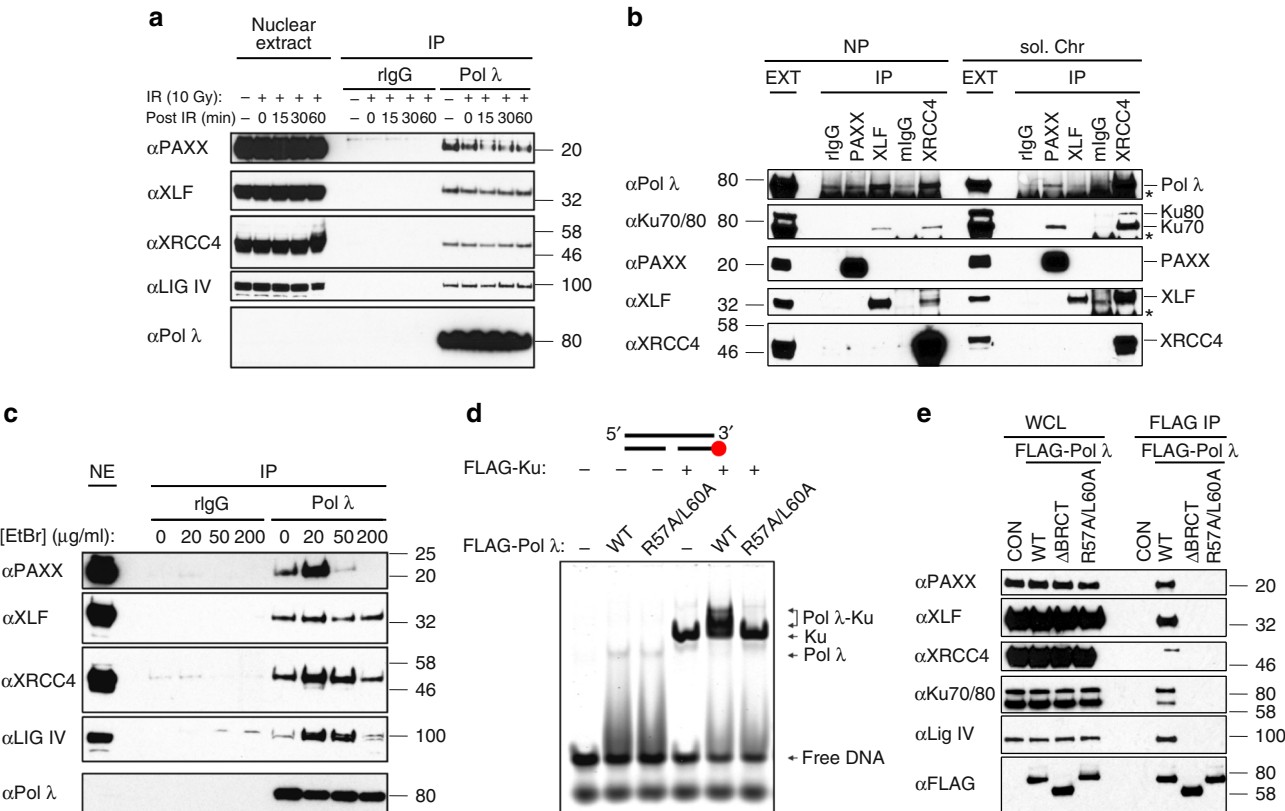

**Fig. 3** Pol λ interacts with XRCC4 family proteins via its BRCT domain in cells. **a** HEK293F cells were irradiated with 10 Gy X-ray or left untreated. Soluble nuclear extracts were isolated following 0–60 min post-irradiation recovery time at 37 °C. Following IP with anti-Pol λ or rabbit IgG (rIgG), Pol λ and associated proteins were resolved by SDS-PAGE and immunoblotted for the indicated NHEJ factors. **b** HEK293F cell nucleoplasmic (NP) or soluble chromatin (sol. Chr) extracts were immunoprecipitated with rIgG, anti-PAXX or -XLF or mouse IgG (mIgG) or anti-XRCC4. Immunoprecipitated proteins were resolved by SDS-PAGE and immunoblotted for the indicated NHEJ factors. **c** As described in Panel A, except that soluble nuclear extracts were incubated with 0-200 μg/ml EtBr for 1 h prior to IP with anti-Pol λ or rIgG. **d** EMSA showing that interaction of Pol λ with DNA-bound Ku requires R57 and L60 in the BRCT domain of Pol λ. Reactions were performed with IRDye® 700-labelled 5nt-gapped dsDNA (33-mer) in the presence or absence of FLAG-Ku70/80 (20 nM) and either FLAG-Pol λ-WT or a R57A/L60A mutant (50 nM). **e** HEK293F cells were transiently transfected with either pCMX-LacZ (control) or pCMX-FLAG-Pol λ-WT, -ΔBRCT or a R57A/L60A mutant and anti-FLAG IPs performed

and Pol λ (Fig. 4d). Deletion of the XLF C-terminal 66 residues resulted in loss of supershifted XLF-Ku-DNA and Pol λ-Ku70/80-DNA-XLF complexes (Fig. 4d), demonstrating that XLF interaction with Pol λ also shows Ku-dependency.

**PAXX and paralogs facilitate recruitment of Pol λ to DSBs.** Pol λ has been shown to localise to DSB sites[27], but the NHEJ factors required for Pol λ recruitment to DSBs in live cells have not been identified. Therefore, we stably expressed EGFP or mCherry fused to the N-termini of Pol λ as a 100 kDa nuclear-localised protein in cells (Fig. 5a, Supplementary Fig. 4a). N-EGFP-Pol λ fusion protein rapidly relocalised to microirradiation-induced DSB sites following UV laser-induced DNA damage (Fig. 5b, c). To examine the role of XRCC4 family members in recruitment/ retention of Pol λ at microirradiation-induced DSBs, PAXX, XLF and XRCC4 KO U2OS cells were generated (Fig. 5d, Supplementary Table 5). We noted that XRCC4 KO cells did not express Lig IV (Fig. 5d)[28]. Loss of PAXX or its paralogs had no effect on nuclear localisation of N-terminal mCherry-Pol λ (Fig. 5e). However, in contrast to rapid relocalisation of Pol λ to laser-induced DSB sites observed in WT cells, Pol λ recruitment to laser-induced DSBs was substantially diminished in PAXX KO cells (Fig. 5f). Pol λ recruitment to DNA lesions was also ablated in both XLF- and XRCC4-deficient cells with defective initial

recruitment similar compared to PAXX KO cells (Fig. 5f, Supplementary Fig. 4b). As PAXX has been shown to promote maximal Ku70 recruitment to DSBs in live murine cells[14], we tested whether the defect in Pol λ recruitment to laser-induced DSBs in human PAXX KO cells correlated with a similar defect in Ku70 relocalisation. N-terminal EGFP-FLAG-Ku70 relocalised to sites of laser-induced DSBs in WT and PAXX KO cells, with no significant difference in Ku70 recruitment observed following laser-induced DSBs (Fig. 5g, Supplementary Fig. 4c). These results suggest PAXX is critically important for recruitment of Pol λ to laser-induced DSBs without exerting a broader effect on Ku retention at DSBs in human cells.

**XRCC4 family proteins stimulate Pol λ gap-filling activity.** As XRCC4 family proteins interact with Pol λ, we next tested whether these NHEJ proteins stimulate Pol λ activity. An assay was developed to measure template-dependent Pol λ activity on a 5nt gapped 33 bp dsDNA substrate in vitro (Supplementary Table 1) [15,16]. Pol λ-WT catalysed gap filling synthesis in a concentration-dependent manner, whereas a catalytically inactive Pol λ mutant (D427A/D429A/D490A; 3D) did not (Fig. 6a, Supplementary Fig. 5a). In addition, Pol λ-WT showed reduced activity towards substrate lacking a 5′-phosphate moiety, consistent with its reported lower binding affinity towards non-phosphorylated

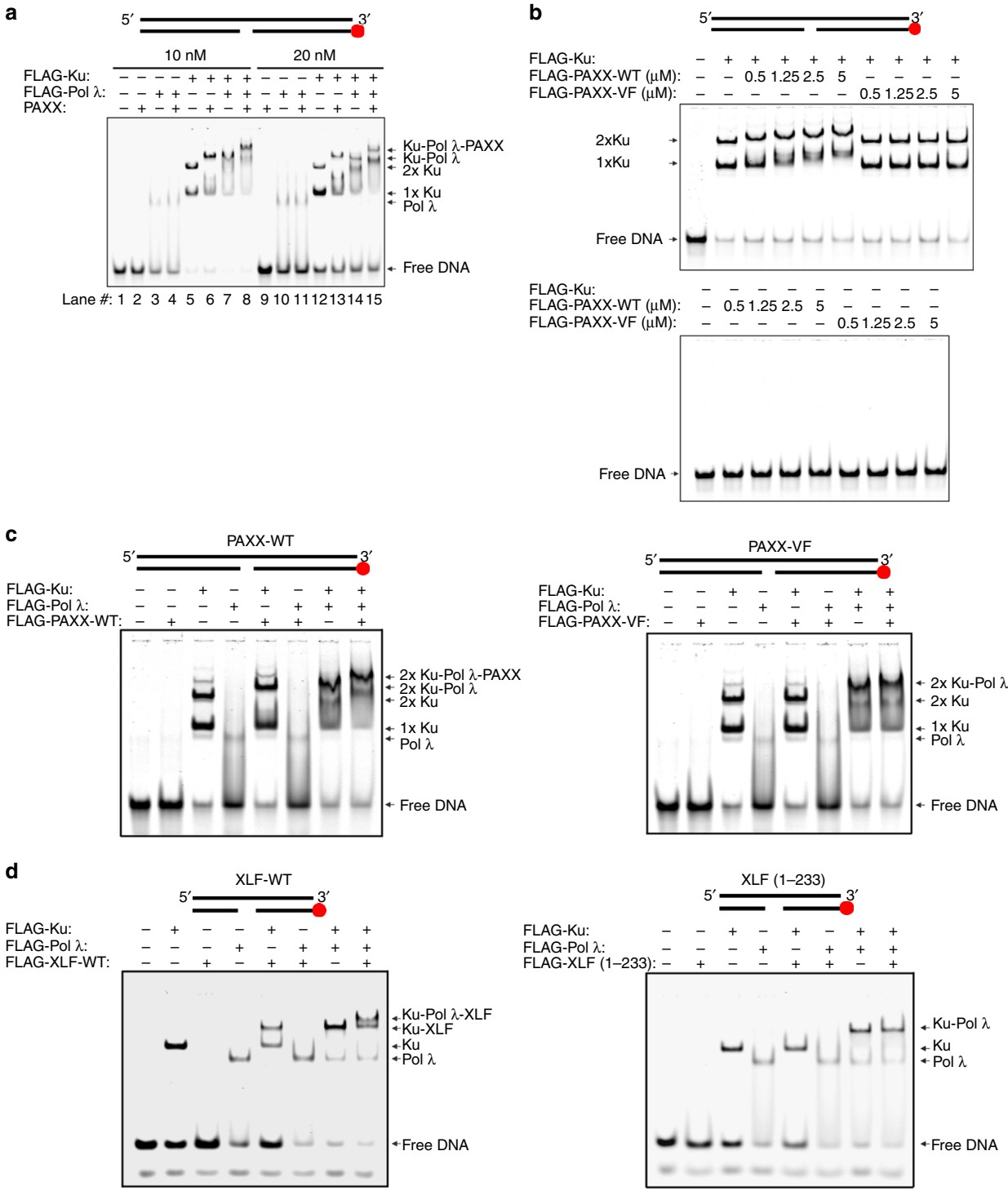

**Fig. 4** Interaction of PAXX and XLF with Pol λ requires C-terminal Ku-binding regions. **a** EMSA showing that interaction of PAXX with Pol λ requires DNA-bound Ku. Reactions were performed with 10 or 20 nM IRDye® 700-labelled 5nt-gapped dsDNA (90-mer) and the following concentrations of FLAG-Ku70/80 (20 nM), FLAG-Pol λ (40 nM) or cleaved PAXX (100 nM). **b** Binding of PAXX to DNA-bound Ku requires C-terminal residues V199 and F201 of PAXX. Reactions were performed with 20 nM IRDye® 700-labelled 5nt-gapped dsDNA (90-mer) and the indicated concentrations of FLAG-PAXX-WT or a FLAG-PAXX-V199A/F201A mutant and FLAG-Ku70/80 (20 nM). **c** As described in Panel A, except that reactions contained FLAG-Ku (20 nM), FLAG-Pol λ (100 nM), FLAG-PAXX-WT (2.5 μM, left panel) or a V199A/F201A mutant (2.5 μM, right panel). **d** As described in Panel A, except that reactions contained FLAG-Ku (20 nM), FLAG-Pol λ (200 nM), FLAG-XLF-WT (2.5 μM, left panel) or a C-terminal FLAG-XLF (aa1-233) deletion mutant (2.5 μM, right panel)

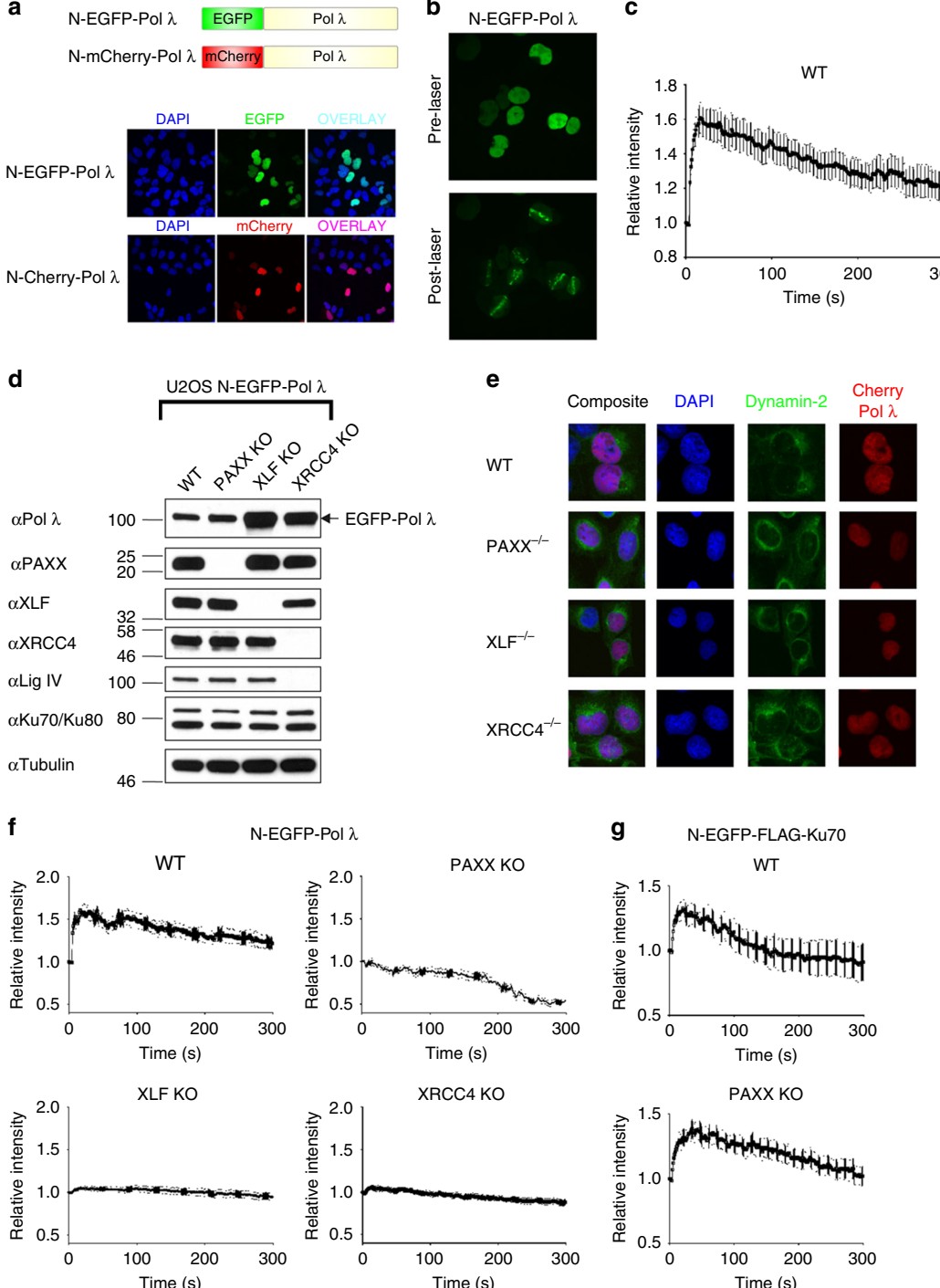

**Fig. 5** Role of XRCC4 family members in the recruitment of Pol λ to laser microirradiation-induced DNA damage sites. **a** Upper, Schematic figure showing N-terminal EGFP- and mCherry-Pol λ fusion proteins; Lower, Representative immunofluorescence images showing that N-terminal EGFP- and mCherry-Pol λ fusion proteins are localised to nuclei in U2OS cells. **b** Recruitment of N-terminal EGFP-Pol λ to laser-induced DNA damage sites in U2OS cells. **c** Time course of N-EGFP-Pol λ recruitment to laser-induced DNA damage sites in U2OS cells. Data shown are the mean and SEM from 16 individual cells. **d** Immunoblot analysis of N-EGFP-Pol λ expressing U2OS cells deficient in PAXX, XLF or XRCC4 generated by CRISPR-Cas9. WCL were resolved by SDS-PAGE and the indicated proteins detected by immunoblotting. **e** Localisation of N-mCherry-Pol λ in U2OS WT, PAXX-, XLF- and XRCC4-deficient cells. Representative immunofluorescence images showing that N-terminal mCherry-Pol λ fusion protein localises to nuclei in PAXX-, XLF- or XRCC4-deficient U2OS cells. Cells were co-stained with DAPI or dynamin-2, a perinuclear-enriched protein. **f** Time course of N-EGFP-Pol λ recruitment to laser-induced DNA damage sites in U2OS-WT cells and cells deficient in PAXX, XLF or XRCC4. Data shown are the mean and SEM from WT, PAXX, XLF and XRCC4 knockout cells. Graphs shown are for the following cell numbers: WT: 8 cells; PAXX KO, 10 cells; XLF KO 17 cells; XRCC4 KO 18 cells. **g** Time course of N-EGFP-FLAG-Ku70 recruitment to laser-induced DNA damage sites in U2OS-WT cells and PAXX KO cells. Data shown are the mean and SEM from WT and PAXX knockout cells. Graphs shown are for the following cell numbers: WT: 17 cells; PAXX KO, 19 cells

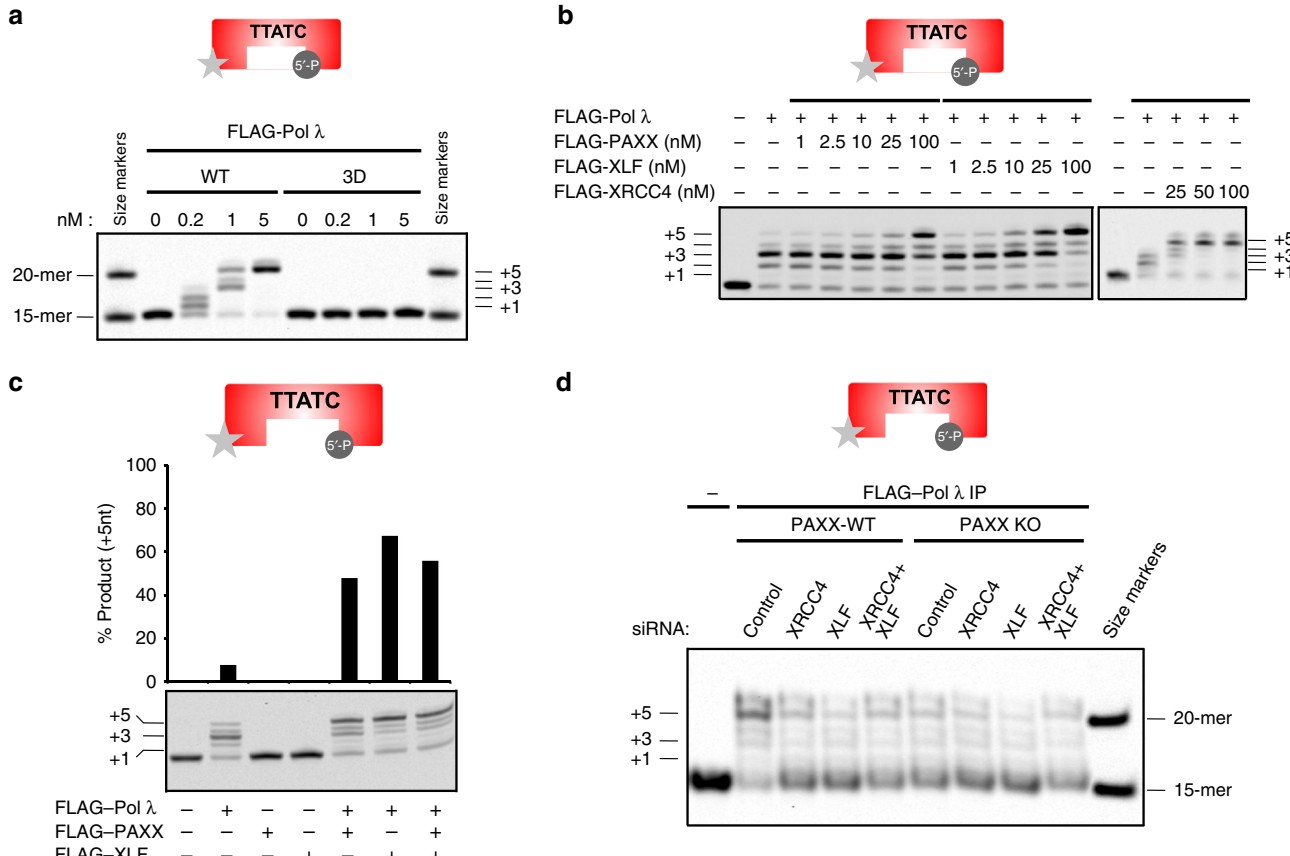

**Fig. 6** XRCC4 family proteins stimulate gap filling synthesis activity of Pol λ. **a** Gap filling activity of Pol λ-WT and a catalytically inactive Pol λ-D427A/D429A/D490A mutant. **b** PAXX, XLF and XRCC4 stimulate gap-filling synthesis activity of Pol λ with an IRDye® 700-labelled 5nt-gapped dsDNA (33-mer) substrate. **c** As described in Panel B, except that some reactions also contained either FLAG-PAXX or –XLF alone **d** Gap filling synthesis assays were performed as described in Panel B with Pol λ immunoprecipitated from RPE-1 PAXX⁺/⁺ or PAXX KO cells incubated with or without XLF or XRCC4 siRNA

substrates (Supplementary Fig. 5b)[25]. Importantly, addition of PAXX or its paralogs strongly enhanced Pol λ-dependent gap filling synthesis in a concentration-dependent manner up to 10-fold (Fig. 6b, c). Addition of PAXX and XLF together did not further stimulate Pol λ-dependent gap filling activity, suggesting that these proteins perform the same function (Fig. 6c). Neither PAXX or XLF exhibited detectable gap filling activity themselves, excluding the possibility these purified proteins contained residual Pol λ activity (Fig. 6c). Next, we evaluated the effect of Ku on PAXX-stimulated Pol λ-mediated gap filling synthesis. Ku had a minor effect on Pol λ gap filling activity. Importantly, PAXX-dependent Pol λ stimulation was observed in the presence of Ku (Supplementary Fig. 5c). Next, we examined a role for XRCC4 family proteins in regulating Pol λ-dependent gap filling activity derived from cell extracts in vitro. Pol λ was immunoprecipitated from RPE-1-WT or PAXX KO cells following depletion of XLF and/or XRCC4 and immunoprecipitates incubated with gapped dsDNA substrate to assess Pol λ activity. Loss of PAXX or depletion of either XLF or XRCC4 resulted in reduced gap filling activity in Pol λ IPs, demonstrating a role for PAXX paralogs in promoting Pol λ activity (Fig. 6d).

**PAXX family proteins stimulate Pol λ via their head domains.** XRCC4 family proteins consist of similarly arranged structural domains with a N-terminal globular head domain followed by a coiled-coil (CC) domain and less conserved C-terminal regions (CTR) (Fig. 7a). The head domain of PAXX more strongly stimulated gap filling activity of Pol λ relative to full length PAXX

(Fig. 7b). Conversely, the CC-CTR region inhibited Pol λ-dependent gap filling activity in a concentration-dependent manner (Fig. 7b). Similar to the PAXX head domain, XLF and XRCC4 head domains also enhanced Pol λ-dependent gap filling activity (Fig. 7c, d). Based on these results we hypothesised that XRCC4 family proteins may also interact via a weak protein–protein interaction with Pol λ via their N-terminal head domains. Far-western blotting using purified proteins showed that the head domains of PAXX, XLF and XRCC4 interact with FLAG-tagged Pol λ (Fig. 7e).

Previous results with the N- and C-terminal regions of PAXX infer that XRCC4 family proteins are not required to bind Ku in order to promote Pol λ-directed gap filling synthesis (Fig. 7b). Indeed, a full-length mutant PAXX protein defective in interaction with DNA-bound Ku, PAXX-VF, enhanced Pol λ-dependent gap filling synthesis comparable to PAXX-WT (Figs 4b, 7f). Deletion of the C-terminal 91 amino acids of GST-tagged PAXX or the C-terminal 59 amino acids from FLAG-PAXX also had little effect on its ability to promote Pol λ-dependent gap filling activity (Supplementary Fig. 6a, b). Similarly, a C-terminal XLF deletion mutant lacking its CTR (XLF (1-233)), including the Ku binding motif, also maintained its ability to enhance gap filling synthesis (Supplementary Fig. 6c). To further exclude a role for Ku binding to either Pol λ or PAXX in stimulating Pol λ-directed gap filling synthesis by PAXX, we compared the ability of PAXX head domain to stimulate gap filling synthesis by either Pol λ-WT or a RL mutant, which does not interact with Ku in vitro or in cells (Fig. 3d, e). Pol λ-WT and a –RL mutant were similarly

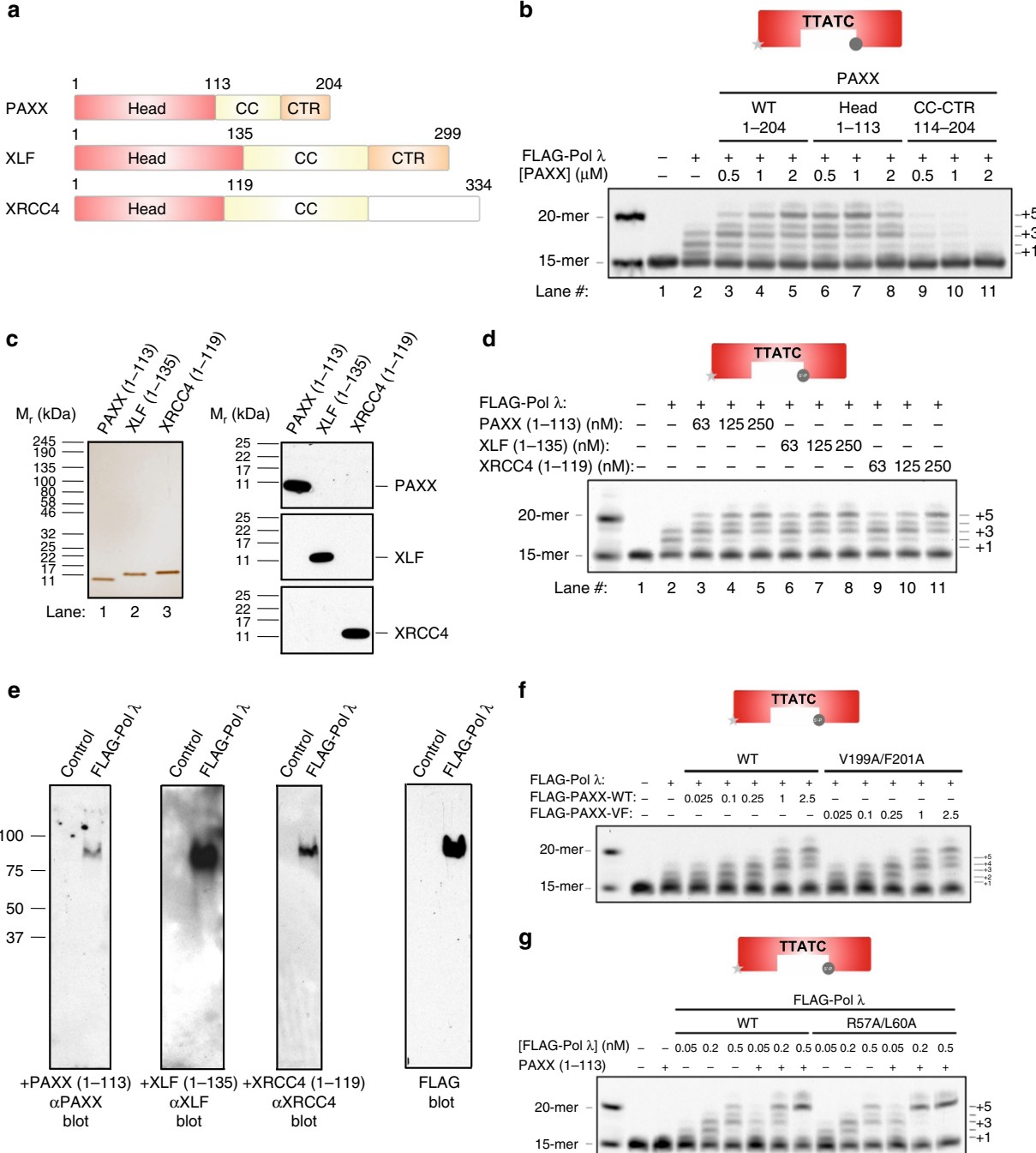

**Fig. 7** Head domains of XRCC4 Family Proteins interact with and stimulate Pol λ-dependent gap filling synthesis. **a** Schematic representation of XRCC4 family proteins. **b** PAXX head domain, but not the CC-CTR region, stimulates gap filling synthesis activity of Pol λ. **c** Silver stain and immunoblot analysis of purified XRCC4 family protein head domains. **d** Head domain of XRCC4 family proteins stimulate Pol λ-dependent gap filling synthesis activity. **e** Far-Western blot analysis showing that Pol λ-WT interacts with the head domain of XRCC4 family proteins. **f** Stimulation of Pol λ-dependent gap filling synthesis activity by PAXX-WT but not PAXX-VF, a non-Ku binding C-terminal mutant. **g** PAXX head domain stimulates comparable gap filling synthesis activity of Pol λ-WT and –R57A/L60A

stimulated by PAXX (1-113), further excluding a role for Ku binding in stimulation of Pol λ-directed gap filling by PAXX (Fig. 7g).

To identify functional domains of Pol λ which mediate the stimulatory effect of XRCC4 family proteins on Pol λ-mediated gap filling synthesis, we generated a panel of N-terminal Pol λ deletion mutants lacking functional domains (BRCT, Serine-Proline-rich (Ser-Pro) and 8 kDa domains; (Fig. 8a,

Supplementary Fig. 7a). First, we assessed the effect of deleting successive N-terminal domains on Pol λ-dependent gap filling synthesis activity. Deletion of the BRCT domain had no effect on Pol λ-dependent gap filling synthesis activity, whereas combined loss of BRCT and Ser-Pro domains moderately enhanced Pol λ-dependent gap filling activity compared to Pol λ-WT (Supplementary Fig. 7b)[29]. Further deletion of the Pol λ 8 kDa domain severely limited its gap filling activity, consistent with its known

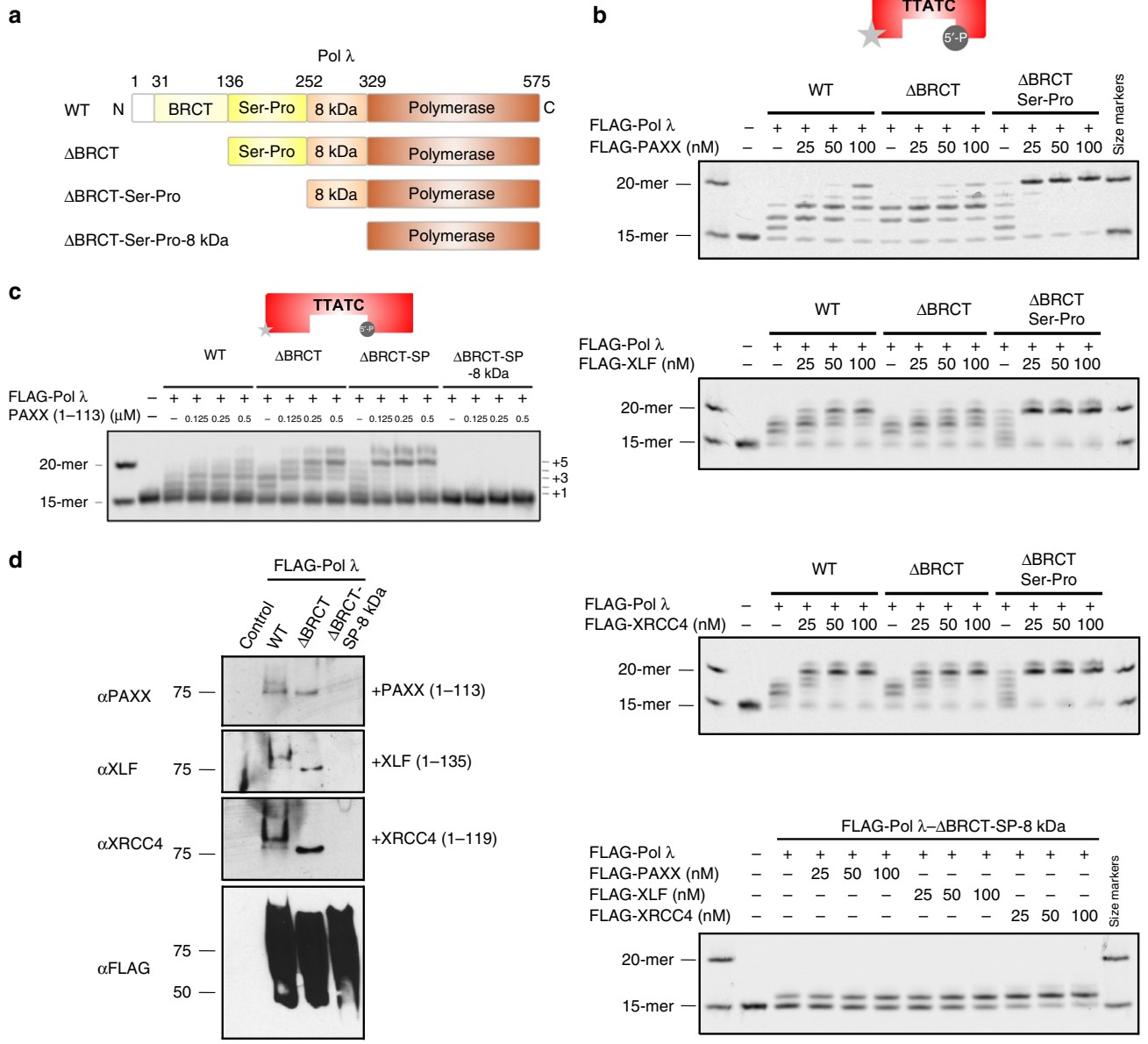

**Fig. 8** The Pol λ 8 kDa domain is required for stimulation of Pol λ-dependent gap filling activity via interaction with the head domain of XRCC4 family proteins. **a** Schematic representation of N-terminal Pol λ deletion mutants. **b** PAXX, XLF and XRCC4 stimulate gap filling synthesis activity of ΔBRCT- and ΔBRCT-Ser-Pro-Pol λ but not ΔBRCT-Ser-Pro-8kDa-Pol λ with an IRDye® 700-labelled 5nt-gapped dsDNA (33-mer) substrate. **c** PAXX head domain promotes gap filling synthesis activity of ΔBRCT- and ΔBRCT-Ser-Pro-Pol λ but not ΔBRCT-Ser-Pro-8kDa-Pol λ. **d** Far-Western blot analysis showing that the head domain of XRCC4 family proteins interacts with Pol λ-WT and -ΔBRCT but not -ΔBRCT-Ser-Pro-8kDa

role in binding 5′ phosphate ends of the DNA gap (Supplementary Fig. 7b)[30,31]. Loss of the BRCT domain had no effect on the ability of XRCC4 family proteins to stimulate Pol λ-dependent gap filling activity, whereas additional deletion of the Ser-Pro domain surprisingly further increased the stimulatory effects of PAXX, XLF and XRCC4 (Fig. 8b). Importantly, removal of the 8 kDa domain resulted in severe loss of responsiveness of Pol λ to XRCC4 family proteins (Fig. 8b, bottom panel). Next, we tested the effect of the PAXX head domain on the N-terminal Pol λ deletion mutants. Similar to full length PAXX, its head domain strongly stimulated gap filling synthesis by Pol λ-WT, -ΔBRCT- and -ΔBRCT-Ser/Pro but not -ΔBRCT-SP-8 kDa, comparable to full length PAXX (Fig. 8c). Consistent with these results, the head domain of PAXX, XLF and XRCC4 interacted with FLAG-Pol λ-WT and -ΔBRCT but not -ΔBRCT-SP-8 kDa (Fig. 8d). Taken

together, our results indicate that PAXX, XLF and XRCC4 share a common ability to enhance Pol λ-mediated gap filling activity. This stimulatory effect appears to be mediated via protein–protein interaction of their structurally conserved head domains and the 8 kDa domain of Pol λ, a region critically involved in binding the 5′ phosphate end of DNA gaps and stabilising scrunched template intermediates (see below)[31].

Pol μ is a related family X DNA polymerase, which plays an overlapping role in NHEJ DSB repair yet, in contrast to Pol λ, has both template-dependent and -independent polymerase activities[16]. A recent study showed that combined loss of both Pol λ and Pol μ resulted in severe hypersensitivity to IR, whereas loss of single polymerases caused a mild radiosensitive phenotype[17]. Therefore, we also tested the effect of PAXX and its paralogs on Pol μ activity using template-dependent gap filling or NHEJ

substrates or a template-independent NHEJ substrate with FLAG-tagged Pol µ (Supplementary Figs 7c–f). Neither PAXX or XLF influenced Pol µ-dependent gap filling activity towards a 3nt gapped 33 bp dsDNA (Supplementary Fig. 7d, left panel). No detectable gap filling activity of Pol µ towards a longer 5nt gapped 33 bp dsDNA substrate was observed in the absence or presence of PAXX or XLF, consistent with studies showing its activity towards substrates with longer gaps was strongly reduced (Supplementary Fig. 7d, right panel; Supplementary Fig. 7e)[32]. PAXX also had no effect on Pol µ activity towards either template-dependent (1 bp) or template-independent (0 bp) NHEJ substrates, whereas higher concentrations of XLF and in particular, XRCC4, appeared to inhibit Pol µ activity with either NHEJ substrate (Supplementary Fig. 7f).

**PAXX and XLF function with Pol λ to ligate noncohesive ends**. Previous studies have demonstrated that PAXX enhances ligation of cohesive or noncohesive DNA ends in a XLF-dependent or -independent manner[11,12,33]. Since PAXX and XLF interact with Pol λ and stimulate its gap filling activity (Figs 1–8), we decided to examine the effects of PAXX and XLF together with Pol λ on the ligation of DNA substrates with distinct DNA ends using a qPCR ligation assay[34,35]. In contrast to gap filling reactions that measure Pol λ enzymatic activity, joining of all combinations of DNA ends was highly dependent upon Ku70/80 heterodimers (Fig. 9a–e, lanes 10 and 12). In the absence of Pol λ, XLF stimulated joining of blunt ends or 3′ overhangs with blunt DNA ends (Fig. 9a–c, lanes 4 and 8)[36]. In contrast, PAXX in the absence of Pol λ moderately enhanced ligation of only blunt DNA ends (Fig. 9c, lanes 4 and 5)[11,33,37]. XLF and Pol λ together promoted ligation of all combinations of DNA ends tested (Fig. 9a–e, lanes 7-8 and 11). On the other hand, PAXX only moderately increased ligation in the presence of Pol λ (Fig. 9a–e, lanes 5-7). Interestingly, PAXX, and XLF together with Pol λ promoted joining of blunt ends with 2-4 bp 3′ overhangs in a manner dependent upon either PAXX or XLF concentrations (Fig. 9a, b, Supplementary Fig. 8a). Finally, we tested involvement of the catalytic activity of Pol λ in stimulation by PAXX and Pol λ of ligation of 5′ overhangs with blunt DNA ends using Pol λ-WT or a catalytically inactive Pol λ mutant (Pol λ-3D). Pol λ-WT, but not Pol λ-3D, co-operated with PAXX to enhance ligation of blunt and 5′ overhang DNA ends (Fig. 9f, lanes 7 and 9). These results show that catalytic activity of Pol λ is required for co-operative effects of PAXX and Pol λ in stimulating ligation of non-compatible DNA ends.

**Pol λ functions with PAXX and XLF in a common pathway**. To understand how Pol λ interacts genetically with PAXX and XLF, we generated a PAXX/XLF DKO cell line in addition to PAXX and XLF KO cell lines (Fig. 10a, Supplementary Table 5) and subsequently depleted Pol λ by siRNA-mediated knockdown (Fig. 10a). Pol λ depletion in WT cells resulted in weak IR sensitivity, consistent with the ability of Pol µ to compensate for loss of Pol λ[17]. PAXX KO cells showed moderate sensitivity to IR, whereas XLF KO cells exhibited stronger IR sensitivity, which was further increased in PAXX/XLF DKO cells (Fig. 10b). These results suggest that PAXX and XLF function in parallel pathways. Pol λ depletion from PAXX KO and to a lesser extent XLF KO cells resulted in increased sensitivity to higher IR doses compared to control cells, suggesting that once one XRCC4 paralog is lost, the NHEJ process becomes more reliant on Pol λ Of note, no significant difference in radiosensitivity was observed following Pol λ depletion in PAXX/XLF DKO cells (Fig. 10b), suggesting that removal of both PAXX and XLF compromises Pol λ activity to a degree that it cannot support effective NHEJ. In summary,

PAXX and XLF non-redundant functions in DSB repair mask a Pol λ-dependent subpathway of NHEJ, in which they operate redundantly.

**Discussion**
PAXX is the most recently identified member of the XRCC4 family of proteins involved in NHEJ DSB repair[10–12]. In this study, we identified DNA-PK holoenzyme as the most abundant PAXX-interacting protein in agreement with another report[37]. Intriguingly, the stoichiometry of individual DNA-PK holoenzyme subunits to PAXX in the soluble chromatin fraction approximated to 1–1.5:1, which underscored direct interaction between PAXX and DNA-bound Ku heterodimers and more specifically, the Ku70 subunit[11,12,37]. Another abundant PAXX-interacting protein which we identified was Pol λ, a member of the family X DNA polymerases. Interaction of Pol λ with both XLF and XRCC4 was also observed. Pol λ together with related Pol µ play important roles to direct DNA synthesis across DSB during NHEJ[6,17]. Interestingly, interaction of Pol λ with PAXX was largely DNA-dependent suggesting possible "bridging" by other DNA-binding factors (Fig. 3c). Indeed, we found DNA-bound complexes of Pol λ−Ku-PAXX and Pol λ−Ku-XLF in vitro (Fig. 4a, c, d). Two related, functionally different Ku-binding motifs were recently identified in multiple DNA damage response proteins including APLF and WRN (APLF-like-Ku binding motif) and XLF, PAXX, WRN (XLF-like motif)[38]. We showed that R57 and L60 located within α-helix 1 of the BRCT domain were required for Pol λ to interact with DNA-bound Ku in vitro (Fig. 3d) and to associate with either PAXX, XLF, XRCC4 or Ku in cells (Fig. 3e). Of note, inspection of the sequence surrounding these residues identified a basic patch followed by a phenylalanine (RxRxxxF), more similar to the newly characterised XLF-like Ku-binding motif[38].

These results led us to hypothesise that PAXX may regulate Pol λ function. Accordingly, PAXX and its paralogs, were required for recruitment of Pol λ to sites of laser-induced DNA damage in live cells. We noticed that Pol λ recruitment to DNA damage sites was defective to a very similar extent in XLF and XRCC4 KO cell lines. As higher-order filament formation is the main common function of these proteins in NHEJ, we hypothesise that defects in such filament accumulation lead to partially impaired Pol λ translocation to DNA lesions. In contrast to this, PAXX KO cells show more profound defects in Pol λ recruitment than XLF or XRCC4 KO cells, which may result from the recently identified ability of PAXX to promote DNA end synapsis together with DNA-PK holoenzyme[39]. Thus, efficient Pol λ recruitment to DNA lesions requires the concerted action of all XRCC4 paralogs. We observed no defect in Ku70 recruitment to DNA damage sites in our PAXX-deficient human cells. These findings suggest that the effect of PAXX on localisation of Pol λ to DSBs is unlikely to be an indirect effect resulting from defective Ku recruitment. In contrast, in murine cells loss of PAXX led to a moderate defect in Ku recruitment to DSB sites, suggesting that in some contexts PAXX may contribute to Ku stability at DNA ends[14]. Of note, human cells express much higher protein levels of Ku heterodimer then rodent cells[40]. This difference might translate into differences in mechanisms of Ku retention at DNA lesions in vivo.

Notably, we show that all XRCC4 members stimulated gap filling activity of Pol λ Previous studies reported that the yeast XLF homolog (Nej1) stimulated gap filling synthesis activity of the yeast family X DNA polymerase Pol4[41] and linked XLF to alignment-based DNA gap filling[42]. Our data extend these findings to show that all human XRCC4 family proteins including PAXX, share a common ability to stimulate Pol λ−dependent gap

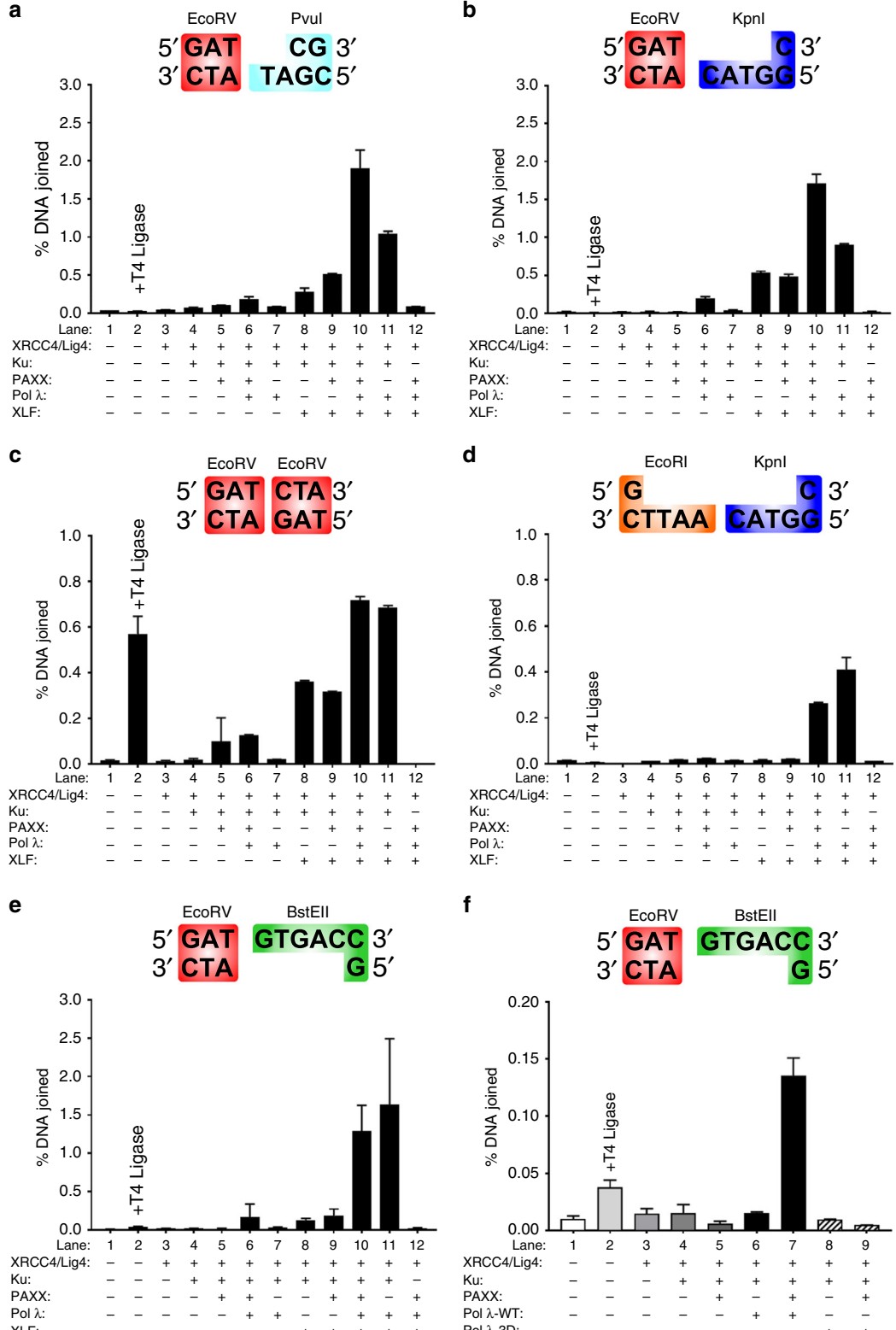

**Fig. 9** PAXX and XLF together with Pol λ to promote ligation of noncohesive DNA ends which requires gap filling activity of Pol λ. **a–e** Linear DNA substrates as shown were incubated with the indicated combinations of XRCC4/Lig IV, Ku70/80, PAXX, XLF and Pol λ and the joining efficiency quantified by qPCR with a TaqMan probe using a standard curve of $\log_{10}$ % joining efficiency versus $C_t$ value generated using prejoined DNA fragments. DNA substrates were as follows: (**a**) EcoRV-PvuI blunt-2nt 3′ overhang; (**b**) EcoRV-KpnI blunt-4nt 3′ overhang; (**c**) EcoRV-EcoRV blunt-blunt ends; (**d**) EcoRI-KpnI 4nt 5′ overhang-4nt 3′ overhang, (**e**) EcoRV-BstEII blunt-5nt 5′ overhang; (**f**) As described in Panel (**e**), except that ligation assays contained either Pol λ-WT or a catalytically inactive Pol λ-3D mutant. Results shown are the mean ± SEM from 2–3 experiments performed in triplicate

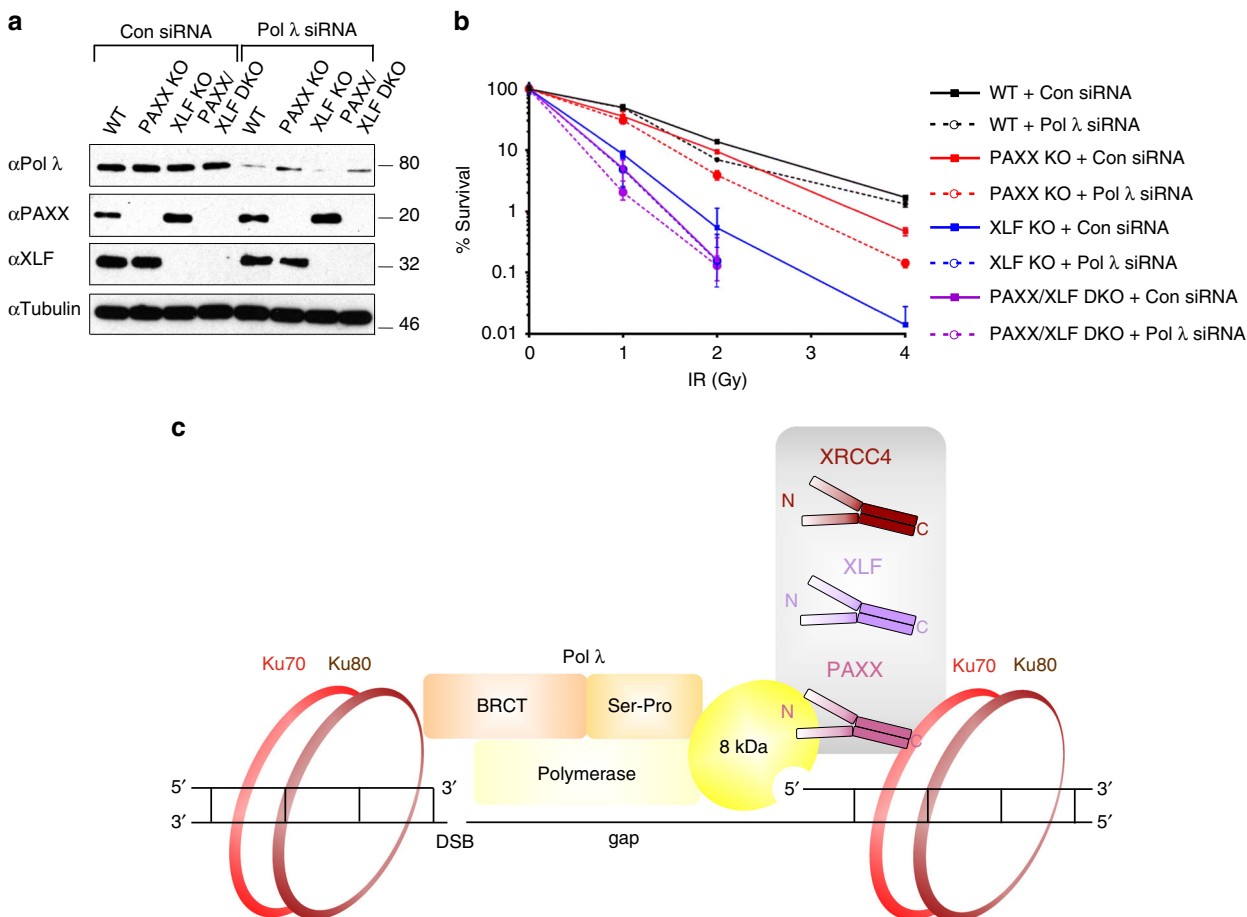

**Fig. 10** Pol λ, PAXX and XLF function in common and parallel pathways. **a** Immunoblot analysis of control- or Pol λ siRNA-depleted U2OS PAXX KO, XLF KO and PAXX/XLF DKO cells. WCL were resolved by SDS-PAGE and indicated proteins detected by immunoblotting. **b** Clonogenic survival assays following IR (0–4 Gy) for U2OS WT, -PAXX KO, -XLF KO and -PAXX/XLF DKO cells with or without depletion of Pol λ Mean and SD from three independent experiments are shown. Statistical analysis was performed using a two-tailed paired t-test to compare cells incubated with Pol λ siRNA with control siRNA: 1 Gy - WT $p = 0.91$, PAXX KO $p = 0.38$, XLF KO $p = 0.10$, PAXX/XLF DKO $p = 0.10$; 2 Gy - WT $p = 0.0003$, PAXX KO $p = 0.0007$, XLF KO $p = 0.36$, PAXX/XLF DKO $p = 0.82$; 4 Gy - WT = 0.02, PAXX KO = 0.002, XLF KO $p =$ not determined, PAXX/XLF DKO $p =$ not determined. **c** Cartoon showing a model for regulation of Pol λ by XRCC4 family proteins. At DSBs that are positioned proximal to a Pol λ substrate gap XRCC4 family proteins strongly interact with Ku heterodimers via their C-terminal regions; their head domains promote gap filling synthesis activity via comparatively weakly binding to the 8 kDa domain of Pol λ, which interacts with the 5′ end of the gap. Pol λ strongly interacts with Ku heterodimers via its N-terminal BRCT domain

filling synthesis activity. We showed that the structurally conserved N-terminal head domain of PAXX, XLF and XRCC4 are sufficient to promote Pol λ-dependent gap filling activity via direct protein-protein interactions. PAXX, XLF and XRCC4 stimulated enzymatic activity of Pol λ by facilitating recognition of the 5′ termini of the substrate gap, as Pol λ lacking the 8 kDa domain responsible for 5′ end recognition lost its responsiveness to XRCC4 family proteins. Furthermore, detectable protein-protein interaction of XRCC4 family proteins occurred between their head domains and the SP-8kDa region of Pol λ. Intriguingly, we observed that PAXX and XLF mutant proteins, which did not bind Ku, retained their ability to fully stimulate Pol λ-dependent gap filling synthesis activity. These findings show that their interaction with Ku is dispensable for stimulating the gap filling enzymatic activity of Pol λ, in contrast to the head domains of XRCC4 family proteins These results are summarised in a cartoon (Fig. 10c) and suggest that XRCC4 family proteins localise on DSBs in the immediate proximity to the gap predominantly via a Ku-dependent mechanism. Weak direct interaction of Pol λ 8 kDa domain with the head domain of

XRCC4 family proteins promotes enzymatic activity of Pol λ to fill DNA gaps. We acknowledge further experiments are required in order to establish the relative contributions of direct and indirect (via binding to Ku) interaction of PAXX (and indeed XLF and XRCC4) with Pol λ to NHEJ in vivo.

In contrast to Pol λ, in vitro gap filling activity of Pol μ was not stimulated by PAXX or XLF. One important distinction between Pol λ and Pol μ which explains their relative abilities to fill longer gaps is that Pol μ includes only two of the three highly conserved residues necessary to form the binding pocket for extra-helical template nucleotides[31]. We hypothesise that XRCC4 family proteins may further stabilise this scrunching pocket within Pol λ. Activity of Pol λ derived from cellular extracts was strongly dependent on each of the XRCC4 paralogs, suggesting that concerted activity of all these proteins is necessary to maintain Pol λ activation under physiological expression levels.

Another finding presented in this report is the demonstration that PAXX, XLF and Pol λ co-operated to efficiently join noncohesive DNA ends. Interestingly, we found that enhanced joining activity between PAXX, XLF and Pol λ was most effective

on small 2–4 bp gaps. On the other hand, XLF co-operated with Pol λ to promote ligation of all combinations of DNA tested. This difference might be a consequence of the smaller size of PAXX in relation to XLF. Furthermore, although PAXX in concert with Pol λ was able to stimulate ligation of substrates with a 5 bp gap, its activity was relatively minor when XLF was added to reactions bearing this substrate. In summary, PAXX collaborated with XLF in gap filling/ligation of gapped DNA substrates in vitro in a manner dependent on the DNA end configuration. Of note, a DNA end-related differential dependence for NHEJ factors was previously observed using an in vitro NHEJ direct ligation system dependent on Artemis[33].

Taken together our study provides novel insights into how XRCC4 family proteins promote joining of noncohesive DNA ends and support the notion that PAXX and XLF have both redundant and non-redundant roles during DNA repair[13,43]. Our study identifies the stimulation of gap synthesis Pol λ activity as a prominent redundant function of these proteins. Given that a recent report found a synergistic contribution of all NHEJ factors to the stability of NHEJ repair complexes[39], it remains possible that the non-redundant PAXX/XLF activities pertain to their unique ability to enhance repair complex formation. In summary, our study identifies the accessory NHEJ factor Pol λ as a key functional mediator of the recently identified NHEJ protein, PAXX and its paralog, XLF.

## Methods

**Cells**. Adherent HEK293H (obtained from Invitrogen) and U2OS (obtained from ATCC) cells were cultured in Dulbecco's Modified Eagle's (DMEM) medium containing 4.5 g/L D-glucose and GlutaMAX (Life Technologies) supplemented with 10% foetal bovine serum. HEK293 cell culture medium also contained pyruvate. Suspension-adapted HEK293 (HEK293F) cells were cultured in Freestyle™ 293 medium (Life Technologies). RPE-1 cells were grown in DMEM/F-12 (1:1) medium containing L-glutamine and 15 mM Hepes supplemented with 10% fetal bovine serum.

**Antibodies**. Antibodies were purchased from the indicated commercial sources: mouse anti-FLAG (M2, F3165, 1:1000-1:10000) and tubulin (Sigma-Aldrich, T6074, 1:1000-1:10000); mouse anti-Ku70 (sc-17789, 1:500), -Ku80 (sc-5280, 1:500), -DNA-PKcs (sc-1832, 1:1000), -Pol μ (sc-398666, 1:200), -PAXX (sc-514359, 1:20-1:100) and –XLF (sc-166488, 1:100-1:200) (all Santa Cruz Biotechnology); rabbit anti-PAXX (Atlas Antibodies, HPA045268, 1:1000), rabbit anti-PAXX (ab126353, 1:1000), Lig IV (ab80514, 1:1000) and dynamin-2 (all Abcam, ab3457, 1:100); rabbit anti-PAXX (D6X7×, CST 92448 S), -XLF (CST 2854 S), -Artemis (CST 13381 S, 1:1000) and -γH2AX (all Cell Signalling Technology, CST 9718 S, 1:1000); mouse anti-XRCC4 (Thermofisher, MA5-24383), rabbit anti-Pol λ (Bethyl, A301-640S, 1:1000-1:10000; A301-641A, 1:1000-1:10000); rabbit anti-XRCC4 (1:500-1:1000) was a generous gift from Dr. Dik Van Gent (Erasmus University, Netherlands).

**Cloning and site-directed mutagenesis of NHEJ factors**. Full-length human XLF, XRCC4, Ku70, Ku80, Lig IV, Pol λ and Pol μ were amplified with Q5 polymerase (NEB) using oligo-dT primed cDNA template prepared from U2OS cells with primers which optionally incorporated additional 5′ sequences encoding either FLAG- or HA tags. PCR products were subcloned into either pCMX eukaryotic expression vector (gift from Dr. Thomas Perlmann, Ludwig Institute for Cancer Research, Sweden) or pGEX-6P-1. pCMX-FLAG-PAXX-V199A/F201A, pCMX-FLAG-Pol λ-R57A/L60A or -D427A/D429A/D490A were generated by site-directed mutagenesis using mutant primers according to the manufacturer's instructions (Agilent Technologies, Stockport, UK)[10]. PAXX, XLF, XRCC4 and Pol λ deletion mutants were generated by PCR using Q5 polymerase using either pCMX-FLAG-PAXX, -XLF, -Pol λ or pGEX-6P-1-PAXX, -XLF, -XRCC4 or Pol λ as templates. pEGFP-C1- and pmCherry-C1-Pol λ were also prepared by PCR using Q5 polymerase using pCMX-FLAG-Pol λ as a template and sense/antisense primers containing XhoI or EcoRI restriction sites, respectively. Sanger sequencing confirmed 100% identity of all insert sequences to respective entries in the NCBI human nucleotide database.

**Transfection of cells**. HEK293F cells were transfected using Freestyle MAX (Life Technologies) according to the manufacturer's instructions or PEI (1.5 μg/ml). Stable HEK293 cell lines expressing FLAG-tagged PAXX, -XLF, and -XRCC4 were generated by co-transfecting mammalian expression vector encoding N-terminal FLAG-tagged proteins with pTKHyg plasmid into 293 H cells (Life Technologies)

and selecting individual clones with hygromycin B (0.2 mg/ml). Individual hygromycin B-resistant clones were screened for expression of full length FLAG-tagged proteins by immunoblotting. A stable clonal HEK293 cell line expressing FLAG-tagged DNA-PKcs was used as described[10]. HEK293F cells were cultured in Freestyle™ 293 medium supplemented with hygromycin B (0.1 mg/ml) between densities of (0.5–3) × 10⁶ cells/ml in conical flasks on a shaking platform (160 rpm) in a humidified 37 °C incubator. RPE-1 cells were transiently transfected using Lipofectamine 2000 according to the manufacturer's instructions (Life Technologies). Stable U2OS cell lines expressing N-terminal tagged EGFP- or mCherry-Pol λ or N-terminal tagged EGFP-FLAG-Ku70 fusion proteins were generated by transfection with PEI (2 μg/ml).

**Immunoaffinity purification of FLAG-tagged NHEJ factors**. For each sample, 250 × 10⁶ HEK293F cells stably expressing FLAG-PAXX, -XLF, -XRCC4, -DNA-PKcs or empty vector (negative control) or transiently expressing FLAG-Pol λ were pelleted by centrifugation at 300 × g. All subsequent procedures were performed on ice or at 4 °C unless indicated. Cells were washed twice in ice-cold PBS-MC (PBS, 1 mM MgCl₂, 1 mM CaCl₂) and gently resuspended in 4.5 ml ice-cold Hypotonic Buffer (10 mM Hepes, pH 7.9, 10 mM KCl, 0.1 mM EDTA, 0.1 mM EGTA supplemented with complete Mini™ protease inhibitor mixture tablets (Roche Diagnostics), 10 mM NaF, 1 mM Na₃VO₄, 10 μM MG132, 1 mM DTT, 1 mM PMSF). After incubation for 15 min, cells were vortexed for 10 s and centrifuged at 2300 × g for 5 min. Crude nuclei were washed with 1 ml Hypotonic Buffer and re-centrifuged as above. Pellets were resuspended and mixed by end-to-end rotation with 5 ml Solubilisation Buffer (20 mM Hepes, pH 7.9, 140 mM NaCl, 0.5 mM MgCl₂, 20% glycerol, 0.5% Igepal CA630 supplemented as described above for 60 min. Following centrifugation at 15000 × g for 30 min, detergent-soluble nuclear extracts were incubated overnight by end-to-end mixing with 25 μl packed low pH glycine-prewashed anti-FLAG M2 agarose (Sigma Aldrich). Detergent-insoluble pellets were washed twice with nuclease incubation buffer (20 mM Hepes, pH 7.9, 1 mM MgCl₂, 20% glycerol) and incubated with benzonase (150 U/ml) for 2 h at 15 °C with gentle mixing followed by addition of 10 mM EDTA. Following centrifugation at 15000 × g for 30 min and readdition of 0.5% Igepal CA630, samples were incubated overnight by end-to-end mixing with 25 μl packed low pH glycine-washed anti-FLAG M2 agarose (Sigma Aldrich). Beads were washed five times with 20 mM Hepes, pH 7.9, 140 mM NaCl, 0.5% Igepal CA630, 0.5 mM MgCl₂, 20% glycerol, 10 mM NaF, 1 mM Na₃VO₄, 10 μM MG132, 1 mM DTT, 1 mM PMSF and proteins eluted with 50 μl Wash buffer containing 3X FLAG peptide (0.2 mg/ml). In some experiments ethidium bromide was added to incrementally reduce protein-DNA association (5-200 μg/ml). Eluates were resolved by SDS-PAGE and gels visualised with either silver stain (Pierce) or for mass spectrometry, stained with colloidal Coomassie (National Diagnostics).

**Identification of interacting proteins by mass spectrometry**. NHEJ factor-interacting proteins were identified by LC-MS/MS as described previously for DNA-PKcs-associated proteins, except that UniProt Human reviewed database (UniProt KB release 2015-04, 20,204 entries) was used as a ref[10]. Raw data files were also analysed using PLGS version 3 and IsoQuant and the same data used for "top 3" absolute quantification of proteins[44,45]. For database searching in PLGS, peptide mass tolerance and fragment mass tolerance were set to auto, one missed cleavage and variable modification for methionine oxidation. False discovery rates (FDR) of 1% and 0.1% were used for PLGS and IsoQuant respectively with only the three most abundant unmodified peptides used for quantification. Data were also analysed using Scaffold version 3.3.1 software (Proteome Software Inc.) as described[10].

**Purification of FLAG- and HA-tagged NHEJ proteins**. HEK293F cells were transiently transfected with FLAG- or HA-tagged constructs for 48-72 h. NHEJ proteins were isolated as described[10], except that following preparation of a high salt (0.42 M NaCl) soluble nuclear extract, NaCl (2 M) was added to a final concentration of 0.6 M to disrupt ionic interactions and 0.5% Igepal CA-630 also added. Soluble 0.6 M NaCl nuclear extracts were mixed with prewashed anti-FLAG M2 agarose or EZ view red anti-HA affinity gel (Sigma) for 3 h at 4 °C prior to elution with 3X FLAG peptide (0.2 mg/ml) or HA peptide (0.1 mg/ml). Specific proteins (FLAG-Ku70/Ku80, FLAG-Lig IV/XRCC4) were further purified by gel filtration chromatography using a Superdex 200 column. Purified FLAG-tagged NHEJ proteins were dialysed overnight with 20 mM Hepes, pH 7.9, 140 mM NaCl, 0.5 mM MgCl₂, 20% glycerol, 1 mM DTT, divided into aliquots and stored at −80 °C.

**Expression and purification of NHEJ proteins in E. Coli**. pGEX-6P-1 constructs were transformed into competent Rosetta2 cells and grown in LB media supplemented with Ampicillin (50 μg/ml) and Chloramphenicol (17 μg/ml) at 37 °C. Cultures were induced with 0.1 mM IPTG at 0.8–1.0 OD₆₀₀ and grown overnight at 16 °C. Following centrifugation, bacterial cell pellets were resuspended in lysis buffer (50 mM Tris pH 7.5, 150 mM NaCl, 1% Triton X-100, 0.1% 2-mercaptoethanol, 0.4 mM PMSF, 1X protease inhibitor cocktail (Roche), 1 mM benzamidine, 1 mM EDTA, 1 mM EGTA) and lysed by sonication. Cell lysates were centrifuged at 20000 × g at 4 °C and supernatants were bound to prewashed

glutathione Sepharose beads for 1 h at RT or 3 h at 4 °C. Beads were washed in batch mode with Wash buffer (50 mM Tris, 0.5 M NaCl, 1% Triton X-100, 0.1% 2-mercaptoethanol, 0.4 mM PMSF, 0.5X protease inhibitor cocktail (Roche), 1 mM benzamidine, 1 mM EDTA, 1 mM EGTA) at 4 °C. Uncleaved GST fusion proteins were eluted in column buffer (20 mM HEPES pH 7.0, 150 mM NaCl, 5% Sucrose, 0.1% CHAPS, 5 mM DTT) supplemented with 20 mM reduced glutathione. Alternatively, column buffer was supplemented with 15U PreScission Protease (GE Healthcare) and incubated at 4 °C overnight to cleave the GST-tag. Eluted proteins were further purified by size exclusion chromatography using either Superdex 75 or 200 columns, which were selected on the basis of the expected protein size and ran on an Äkta protein purification microsystem (GE Healthcare).

**Electrophoretic mobility shift (EMSA) assays**. dsDNA (33- or 90-bp) containing either a 1-, 3- or 5nt gap were generated by annealing the indicated oligonucleotides (oligonucleotides used in this study are listed in Supplementary Table 1), one strand of which was labelled at the 5′ terminus with an IRDye® 700 (IDT, Coralville, IN, USA). Binding reactions (10 µl) were performed by incubating 10–20 nM IR700-labelled dsDNA with the indicated concentrations of purified proteins in 50 mM TrisHCl pH7.5, 100 mM KCl, 2.5 mM MgCl$_2$, 1 mM DTT, 4% glycerol, 0.05% Triton X-100, 0.1 mg/ml BSA for 30 min at 25 °C in the dark and resolved at 50 V on 4-5% native polyacrylamide gels in 0.5 x TBE at 4 °C in the dark. Free and protein-bound IR700-labelled dsDNA were visualised using a LICOR Odyssey CLx imaging system (LICOR, Cambridge, UK).

**DNA polymerase-mediated gap filling and NHEJ assays**. PAGE-purified DNA oligonucleotides were purchased from IDT (Coralville, IN, USA). Gapped and NHEJ substrates were synthesised and annealed as described, except that 5′-termini of specific oligonucleotides to be extended were labelled with IRDye® 700[25]. DNA polymerase reactions (20 µl) were performed at 30 °C for 30 min in 50 mM Tris pH 7.5, 4% Glycerol, 2.5 mM MgCl$_2$, 1 mM DTT, 0.1 mg/ml BSA, 1 µM dNTPs with 5 nM IRDye®-700-labelled gapped/NHEJ DNA substrates and the indicated concentration of FLAG-tagged Pol λ and Pol µ purified from HEK293 cells or GST-Pol λ expressed and purified from *E. Coli*. Where indicated some assays were also performed in the presence of 10 nM FLAG-Ku and 1 or 10 nM unlabelled dsDNA (0.4kB). Assays were also performed with endogenous Pol λ immunoprecipitated from RPE-1 cells using anti-rabbit Pol λ (Bethyl A301-640A). Reactions were stopped by addition of an equal volume of loading buffer (1 x TBE, 7 M urea, 10% glycerol, 0.2% Orange G), denatured for 4 min at 70 °C and oligonucleotides resolved using denaturing 7.5 M Urea/20% polyacrylamide gels. IRDye®-800-labelled 15- and 20 nt oligonucleotides were used as size markers. Scanning and analysis was performed using a LICOR Odyssey CLx imaging system (LICOR, Cambridge, UK).

**Clonogenic survival assays**. U2OS-WT, -PAXX KO, -XLF KO or -PAXX/XLF DKO cells were transfected in 6-well plates with control or Pol λ siRNA for 72 h. Following trypsinisation, washed cells were counted, replated in triplicate at densities of 1000–30,000 cells per 10 cm dish and exposed to X-Rays (0–4 Gy) and grown for 10–14 days to form colonies. Colonies were fixed in 75% methanol/25% acetic acid, prior to staining with PBS/0.05% (w/v) crystal violet and counting. The survival fraction was determined from the plating efficiency of the specific IR dose relative to the plating efficiency of non-irradiated controls.

**DNA ligation assays**. A qPCR assay was developed to quantify the joining of two specific DNA ends in a single orientation as described with the following modifications[34,35]. Briefly, pGEX-6P-1 was used as a template to create two DNA fragments; DNA1, an 800 bp fragment (bases 2000-2799) and DNA2, an 850 bp fragment (bases 2900-3749). DNA1 and DNA2 were amplified with Q5 polymerase using the primers listed in Supplementary Table 2 or 3. PCR-amplified DNA1 and DNA2 fragments were gel purified and subcloned into pJET1.2 blunt cloning vector. DNA1 and DNA2 fragments were then released by digestion with the appropriate restriction enzymes and further gel purified to exclude possible contamination due to undigested PCR amplification products. Ligation reactions (20 µl) were prepared in buffer containing 25 mM Tris pH 8.0, 100 mM NaCl, 0.1 mM EDTA, 0.05% Triton X-100, 5% PEG 8000, 2 mM DTT and 0.05 mg/ml BSA. Proteins (2.5 nM XRCC4/Lig IV, 5 nM Ku70/80, 2.5 nM XLF, 10 nM PAXX and 10 nM Pol λ) were preincubated with 1 nM DNA1 and DNA2 for 5 mins at 25 °C. Reactions were started with 25 µM dNTPs, 0.1 mM ATP and 5 mM MgCl$_2$, incubated at 37 °C for 30 min and stopped by addition of 2 µl 0.5 M EDTA, pH 8.0. DNA was subsequently purified using Monarch DNA cleanup columns. qPCR was then used to quantify DNA end joining efficiency as follows: 20 µl qPCR reactions in triplicate were prepared using 1 µl purified end-joined DNA, 10 µl TaqMan reaction buffer, 300 nM forward (CGTGTCTTACCGGGTTG) and reverse (GGAAAGAACATGTGAGCAAA) primers and 100 nM TaqMan probe ([6FAM] AAACGCCAGCAACGCGGCCTTT[3TAMRA]). The primer and probe sequences for TaqMan qPCR were designed using the MacVector software. Prejoined DNA fragments were created to establish a standard curve of log$_{10}$% joining efficiency versus cycle number required for the fluorescent signal to cross the threshold amount of DNA (Ct value). Prejoined DNA (1 nM) was defined as 100% joining efficiency and 10-fold serial dilutions were performed to establish a

standard curve to 0.0001% joining efficiency. Experimental Ct values were then interpolated to the standard curve to obtain the DNA joining efficiency for any given ligation reaction using Graphpad Prism (version 7, Graphpad Software Inc., La Jolla, CA, USA).

**Crosslinking far-western blotting**. Purified FLAG-Pol λ (0.25-1 µM) was electrophoresed on non-denaturing PAGE followed by transfer to nitrocellulose membranes. After overnight blocking with PBS, 0.1% Tween 20 (PBST), 5% milk, membranes were incubated with PAXX/XLF/XRCC4 head domains (0.5 µM) diluted in PBST for 4 h at RT. 1-ethyl-3-(3-dimethylaminopropyl)-carbodiimide (EDC, 0.4 mg/ml) and N-hydroxy-succinimide (NHS, 0.6 mg/ml) were added for the final 30 min to stabilise weak protein-protein interactions and crosslinking quenched with 20 mM 2-mercaptoethanol[46]. After washing 3 x TBST, membranes were incubated with the following antibodies to detect individual head domains: mouse anti-PAXX (Santa Cruz Biotechnology sc-514359), mouse anti-XLF (Santa Cruz Biotechnology sc-166488), rabbit anti-XRCC4 (gift from Dr. Dik Van Gent).

**siRNA-mediated knockdown experiments**. Cells were transfected with 20 nM ON-TARGETplus siRNA SMARTpools (GE Healthcare) using Lipofectamine RNAiMAX (Life Technologies) according to the manufacturer's protocol.

**CRISPR-Cas9-mediated deletion**. U2OS cells were transfected with a CMV promoter-driven Cas9-Puro expression vector and independent predesigned guide RNAs (gRNA; Dharmacon; Supplementary Table 4) for each targeted gene using Lipofectamine 2000. After 24 h at 37 °C, puromycin (4µg/ml) was added for an additional 24 h to select Cas9-expressing cells, and subsequently replaced with DMEM in the absence of puromycin for a further 48 h. Immunoblotting was performed to estimate relative gRNA targeting efficiency and cells transfected with the two highest targeting efficiency gRNAs either plated at a density of 500 cells per 10 cm dish or plated in 96-well plates at a density of 12.5 cells/ml. After 1–2 weeks, separate individual colonies were transferred to 12-well plates using trypsin/EDTA-soaked cloning discs, grown and subsequently analysed by immunoblotting for the absence of PAXX, XLF or XRCC4. Identification of CRISPR-Cas9-mediated mutations near gRNA target sequences was performed by amplification of 0.4–1 kb genomic DNA regions spanning gRNA target sequences using Q5 high fidelity DNA polymerase (Supplementary Tables 4-5). PCR products were ligated into pJET1.2 and DNA sequencing performed on at least six colonies for each independent clonal knockout cell line.

**Immunofluorescence of fixed cells**. Cells were grown on glass coverslips pre-coated with either poly-D-lysine (0.1 mg/ml) or collagen (35 µg/ml). Cells were fixed with 4% paraformaldehyde for 10 min at RT, permeabilised and blocked with PBS, 0.3% Triton X-100, 5%(v/v) goat serum for at least 2 h at 4 °C. Cells were incubated overnight at 4 °C with the indicated primary antibodies and subsequently with secondary antibodies and DAPI (5-10 µg/ml) for 1 h at RT. Images were acquired using a Zeiss LSM510 confocal microscope and analysis performed using the Zen software (Carl Zeiss).

**Laser microirradiation and live cell imaging**. U2OS cell lines stably expressing N-terminal in frame fusions of either EGFP or mCherry with Pol λ were generated by transfection using PEI (0.5 µg/ml) mixed with pEGFP-C1- or pmCherry-C1-Pol λ. After 48 h, selection was performed by addition of G418 (1 mg/ml) to generate a stable polyclonal population of U2OS cells expressing a range of levels of N-EGFP- or -mCherry fusion proteins. Cells were sensitised with Hoechst 33342 (10 µM) immediately prior to live cell imaging. Live cell time lapse imaging combining laser microirradiation with confocal microscopy was performed by capturing images on a Marianas-SDC system from Intelligent Imaging Innovations (3i). This system uses a Yokogawa CSU-W with 150 mW 488 nm excitation and Hamamatsu Flash 4.0 v2+ camera. DNA damage was introduced using the 3i 'Ablate' UV 355 nm pulsed laser system (70 µJ per pulse at 200 Hz) which is focussed to a diffraction limited spot at the sample plane and steered along a user-defined line by the 3i 'VectorM' MEMS mirror scanner and Slidebook software. Relative intensity at laser-damaged sites was calculated as follows: following background subtraction the mean value of the intensity of each damage site at each time point was divided by the mean value of intensity in an undamaged cell to adjust for time-dependent photobleaching during time courses. For each cell, relative intensity was calculated by normalising the intensity at time = 0 s to a value of 1.

**Statistical methods**. Statistical analysis was performed using Graphpad Prism (version 7, Graphpad Software Inc., La Jolla, CA, USA). In Fig. 5c, f, data are the mean EGFP fluorescence (± SEM) in the laser track corrected for photobleaching and normalised to mean EGFP fluorescence before irradiation. An unpaired two-tailed *t*-test was used to analyse data shown in Fig. 5f and is shown in Supplementary Fig. 4b. In Fig. 9, all results shown are the mean ± SEM from 2 to 3 experiments performed in triplicate.

## Data availability

All data supporting the findings of this study are available from the corresponding author on reasonable request. Uncropped western blots are shown in Supplementary Fig. 9A-G. The mass spectrometry proteomics data have been deposited to the ProteomeXchange Consortium (http://proteomecentral.proteomexchange.org) via the PRIDE partner repository with the dataset identifier PXD010891 and 10.6019/PXD010891. All other data are available from the authors upon reasonable request.

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

## Acknowledgements

This work was supported by the Medical Research Council (MRC) UK. We would like to thank Prof. Steve Jackson for providing the RPE-1 PAXX knockout cell line and Dr. Dik Van Gent for the rabbit polyclonal XRCC4 antibody. We would like to thank Dr. Lev Solyakov for help with protein purifications.

## Author contributions

A.C., D.M., R.J-J., G.S., C.L. and M.M. performed experiments; A.C., D.M. and R.J-J. analysed data; A.C., D.M. and M.M. wrote the manuscript with input and editing from all authors; K.C. and M.M. supervised the project.

## Additional information

**Competing interests:** The authors declare no competing interests.

