## [Peer Review File · Nature Communications]

Reviewers' comments:

Reviewer #1 (Remarks to the Author):

Craxton et al characterize a physical interaction between Pol lambda and PAXX, two factors with minor backup roles in NHEJ. The evidence in support of the interaction includes pull-downs, EMSAs, and accumulation at sites of damage in cells, and this evidence is convincing. However, there is no evidence provided that this interaction is important for in vivo repair (e.g. sensitivity to ionizing radiation, or integrated NHEJ substrate). It is unlikely this work will be of wide interest.

Minor comments

- 1) The authors argue In vitro end joining reactions show a larger influence of PAXX than XLF on repair in the presence of Pol lambda, but there was no attempt to determine if this result was simply due to the presence of twice as much PAXX, relative to XLF, in reactions. Both should be titrated to determine if there is any specificity for stimulation of activity by PAXX.
- 2) The cluster diagrams in Figure 1c and 1d are unreadable. The authors should consider showing fewer diagrams that are large enough to be legible, or just refer to the supplementary excel spread sheet.

Reviewer #2 (Remarks to the Author):

In this manuscript entitled "PAXX is essential for directing the activity of DNA Polymerase λ in DNA repair" described a molecular mechanism of PAXX, a recently identified NHEJ protein. They found that PAXX was required for the recruitment and the activation of Pol λ . They showed that PAXX and its paralog XLF synergized with Pol λ to promote joining of incompatible DNA ends.

Overall, this is a very interesting story and contains a lot of high quality data. However, the molecular mechanism is not clear enough. For example, how does PAXX promote the activity of Pol λ ? Which region or point of PAXX is required for its interaction with Pol λ ? The function of Ku in PAXX-activated Pol λ ?

Major points:

1. To address the molecular mechanism, it's necessary to find which points of PAXX and Pol λ are required for their direct interaction. So is XLF.
2. All paralogs of XRCC4 can promote Pol λ activity. What's the difference? Why the manuscript only emphasized PAXX?

Specific points:

1. Overexpression of Flag-tagged protein for IP usually leads to artifact results. It's necessary to do a IP-Western with endogenous antibodies for PAXX, XLF and XRCC4 to confirm their results.
2. Page 18, line 418. "PAXX^{-/-}, 418 XLF^{-/-} and XRCC4^{-/-} U2OS cells", U2OS cells are aneuploidy.
3. Fig.3C, it's worth testing whether XLF also have similar activity.
4. Fig. 5D, the quality of this figure is low.
5. Fig. 5B, G&F. To excluding that the stimulation activity came from contamination, it's necessary to include a point mutant of PAXX, which does not interact with Pol λ , as a negative control.
6. Fig. 6, it's also better to use a point mutant of PAXX as a negative control.

Reviewer #3 (Remarks to the Author):

In this manuscript Craxton et al. dissect the role and interactions of PAXX in the non-homologous end joining (NHEJ) pathway and identify Pol L as a major downstream factor. First, they generate the PAXX interactome in parallel with its paralogs XRCC4 and XLF, confirming association of PAXX with NHEJ factors, including Pol L, preferentially in soluble chromatin and the overlap with XLF and XRCC4. Next, they focus in the interaction of PAXX and DNA polymerase Pol L. Pol L interacts with PAXX and its paralogs and PAXX interaction is dependent of DNA and Ku70/80. Next, using laser-induced DSBs they find that PAXX or its paralogs are required for the recruitment in vivo of Pol L but not for Ku recruitment and retention. When they assay Pol L gap filling activity in vitro they find that either PAXX or its paralogs enhances Pol L activity, and the absence of PAXX or any of its paralogs reduced Pol L activity when Pol L is immunoprecipitated from cells. Using Pol L mutants lacking functional domains they dissect how PAXX and its paralogs enhance the gap-filling activity of Pol L, and in the case of PAXX this is independent of its interaction with Ku. Neither PAXX nor XLF have an effect on Pol Mu activity. Finally, using a qPCR ligation assay with a substrate with distinct DNA ends they found that joining depends on the presence of Ku70/80, and Pol L and PAXX have cooperative effects in stimulating ligation of non-compatible ends.

The paper provides an advance in our understanding of NHEJ, by revealing a new link between PAXX and gap filling activities. The datasets are generally convincing, but the author should try to address the following points:

- what is the role of Ku in the gap filling in vitro experiments? Ku is essential for in vitro interaction of PAXX with Pol L, and essential for the joining in the qPCR ligation assay, but in there is no Ku dependency in the gap-filling experiments and the mutant PAXX-VF enhances Pol L in the same way than the WT. How is possible that PAXX enhances Pol L gap filling in vitro without this interaction if PAXX is dependent on KU? (in the Methods they don't say if Ku is present in the gap filling experiments)

- Which interacting proteins are only associated with PAXX and exclusive of XLF (or visa versa)? Is there any upstream or downstream factor that might point to a non-redundant role between PAXX and XLF during DNA repair

- With the qPCR ligation assay they demonstrate that PAXX synergize with XLF in the joining of non-compatible ends only in the presence of small gaps or blunt ends. They show the % of joining with and without PAXX in the presence of the rest of components of NHEJ. To better compare the role of PAXX and XLF in the joining it would be interesting to assess the % of joining with and without XLF in the presence of the other components of NHEJ.

- Do they have data on the Pol L interactome?

Minor:

- Make 1C-D more clear in the labels, it is difficult to read. Highlight PP6, DYN1-3, TRF2, RAP1.
- Cropped supplementary 2 labels.
- In supplementary figure 3 C, why does the FLAG IP with PAXX-WT not immunoprecipitate any NHEJ component and it is the same as the mutant VF?

Reviewers' comments:

Reviewer #1 (Remarks to the Author):

Craxton et al characterize a physical interaction between Pol lambda and PAXX, two factors with minor backup roles in NHEJ. The evidence in support of the interaction includes pull-downs, EMSAs, and accumulation at sites of damage in cells, and this evidence is convincing. However, there is no evidence provided that this interaction is important for *in vivo* repair (e.g. sensitivity to ionizing radiation, or integrated NHEJ substrate). It is unlikely this work will be of wide interest.

To address the *in vivo* relevance of our study showing that XRCC4 family proteins, including PAXX, regulate Pol λ activity, we performed clonogenic survival assays measuring the sensitivity to ionising radiation, as suggested by the reviewer. Our results show that while PAXX and XLF have some specific roles in protecting cells following ionising radiation, they redundantly control the activity of Pol λ (Fig. 10). The similar radiosensitivities of PAXX/XLF DKO and Pol λ -depleted PAXX/XLF DKO cells suggest that Pol λ lies on the same pathway as both PAXX and XLF to repair DSBs (Fig. 10), consistent with our *in vitro* studies demonstrating that PAXX and XLF jointly promote Pol λ -dependent gap filling activity .

The reviewer refers to PAXX as a factor with minor roles in NHEJ. We would like to point out a recent study by Liu et al. 2017 published in *Nature Communications* that determined the phenotype of PAXX knockout mice (1). While the phenotype of PAXX single knockouts (and XLF single knockouts) is mild (we assume that this observation made the referee conclude the backup role for PAXX), in double PAXX/XLF knockout mice NHEJ is completely dysfunctional. A currently prevailing interpretation of these data is that there is substantial genetic redundancy between several NHEJ factors that translates into minor phenotypes of single mutants (2). A similar point can be made about Pol λ enzyme being compensated by the activity of Pol μ (3).

Minor comments

1) The authors argue In vitro end joining reactions show a larger influence of PAXX than XLF on repair in the presence of Pol lambda, but there was no attempt to determine if this result was simply due to the presence of twice as much PAXX, relative to XLF, in reactions. Both should be titrated to determine if there is any specificity for stimulation of activity by PAXX.

We acknowledge in our original submission that we may have create an impression that “In vitro end joining reactions show a larger influence of PAXX than XLF on repair in the presence of Pol lambda”. Therefore, we have significantly revised text related to these results to clearly state that “In the absence of Pol λ , XLF robustly stimulated joining of blunt ends or 3' overhangs with blunt DNA ends (Fig. 9A-E) (4). In contrast, PAXX in the absence of Pol λ moderately enhanced ligation of blunt DNA ends (EcoRV-EcoRV; Fig. 9C, compare lanes 4 and 5) (5-7). XLF and Pol λ synergistically promoted ligation of all combinations of DNA ends tested (Fig. 9A-E, compare lanes 7, 8 and 11). On the other hand, PAXX only moderately increased ligation in the presence of Pol λ (Fig. 9A-E, compare lanes 5-7). As suggested, we also titrated PAXX and XLF in the presence of other ligation reaction components with a fixed concentration (5 nM) of either PAXX (for titration of XLF) or XLF (for PAXX titrations). Our results show that XLF (Supplementary Fig. 8A, lane 5) stimulates ligation of blunt and 3' overhang ends more than PAXX (Supplementary Fig. 8A, lane 9), although synergistic stimulation of DNA end joining is strongest in the presence of increasing amounts of PAXX (Supplementary Fig. 8A).

2) The cluster diagrams in Figure 1c and 1d are unreadable. The authors should consider showing fewer diagrams that are large enough to be legible, or just refer to the supplementary excel spread sheet.

As suggested, we created 2 separate full size Figures (new Figs. 1 and 2) from the cluster diagrams in original Figs 1C and D and increased text font sizes. Individual protein labels are now legible. In addition, we have also moved original Figs. 1A and 1B to the Supplemental data section (Supplemental Figs. 1C and 1D).

Reviewer #2 (Remarks to the Author):

In this manuscript entitled “PAXX is essential for directing the activity of DNA Polymerase λ in DNA repair” described a molecular mechanism of PAXX, a recently identified NHEJ protein. They found that PAXX was required for the recruitment and the activation of Pol λ . They showed that PAXX and its paralog XLF synergized with Pol λ to promote joining of incompatible DNA ends.

Overall, this is a very interesting story and contains a lot of high quality data. However, the molecular mechanism is not clear enough. For example, how does PAXX promote the activity of Pol λ ?

To address how PAXX stimulates gap-filling synthesis activity of Pol λ , we expressed and purified its head domain (aa1-113) or coiled-coil and C-terminal regions (aa114-204) from *E. Coli* (Figs. 7A and C). These deletion analysis studies of PAXX showed that its N-terminal head domain (aa 1-113) more strongly stimulated Pol λ -dependent gap filling activity compared to full length PAXX (Fig. 7B). In contrast, PAXX (aa114-204) inhibited Pol λ -dependent gap filling activity (Fig. 7B). The head domains of XLF and XRCC4 also enhanced Pol λ -dependent gap filling activity (Fig. 7D). Furthermore, we showed that the head domains from each XRCC4 family protein also interacted with the Ser-Pro-8kDa region of Pol λ (Figs. 7E and 8D). We also extended our studies investigating a possible role for the C-terminal Ku-binding regions of PAXX and XLF in promoting Pol λ -dependent gap filling activity. Using either a 2-amino acid PAXX point mutant (PAXX-V199A/F201A) or C-terminal deletion PAXX and XLF mutants, we showed that their Ku-binding regions are not required for stimulation of Pol λ -dependent gap filling activity (Fig. 7F, Supplemental Fig. 6A-C). In summary, our new results show that the structurally conserved head domain of XRCC4 family proteins share the ability to promote gap-filling by Pol λ via interaction with the Ser-Pro-8kDa of Pol λ . For clarity, we have now included a cartoon model depicting this molecular mechanism (Fig. 10C).

Which region or point of PAXX is required for its interaction with Pol λ ? The function of Ku in PAXX-activated Pol λ ?

Major points:

1. To address the molecular mechanism, it's necessary to find which points of PAXX and Pol λ are required for their direct interaction. So is XLF.

As stated in our response above and in our revised manuscript, we have shown that the head domain of XRCC4 family proteins is required to stimulate Pol λ -dependent gap filling activity (Fig. 7B and D, Fig. 8C). To establish whether the head domain of XRCC4 family proteins also interacted directly with Pol λ , we used crosslinking far western blotting (8). Our results demonstrate that the head domain of each XRCC4 family protein associated with the Ser-Pro-8kDa region of Pol λ (Figs. 7E and 8D). These results are summarised in a cartoon shown in Fig. 10C, which illustrates that Pol λ interacts with PAXX and XLF via at least two mechanisms; principally via an indirect, DNA-bound Ku-dependent mechanism which requires the C-terminal region of XRCC4 family proteins and the BRCT domain of Pol λ . Importantly, we have also revealed a second, albeit weaker, direct interaction. This involves the head domains of XRCC4 family proteins and the Ser-Pro-8kDa region of Pol λ , which facilitates enhanced Pol λ -dependent gap filling activity.

2. All paralogs of XRCC4 can promote Pol λ activity. What's the difference? Why the manuscript only emphasized PAXX?

We concur with the Reviewer's comment regarding the strong emphasis we placed on PAXX in our original submission, which in part reflected its discovery as the most recently discovered XRCC4 family member. Consequently, in our revised manuscript we have now widened our conclusions to reflect the broader significance of our results which show that all XRCC4 family proteins stimulate Pol λ -dependent gap filling activity and also are required for Pol λ recruitment to DNA damage sites. This is reflected in the new title "**PAXX, XLF and XRCC4 synergistically direct the activity of DNA polymerase λ in DNA repair**", the abstract and results sections. In addition, we show that the structurally conserved head domain of all XRCC4 family protein associates with and enhances Pol λ -dependent gap filling activity

(Figs. 7 and 8). Additional studies have also been performed with XLF to complement results shown for PAXX (Fig. 4D).

Specific points:

1. Overexpression of Flag-tagged protein for IP usually leads to artifact results. It's necessary to do a IP-Western with endogenous antibodies for PAXX, XLF and XRCC4 to confirm their results.

In our original submission, IP of endogenous Pol λ co-immunoprecipitated each XRCC4 family protein (PAXX, XLF and XRCC4) (Figs. 3A and C). In our revised manuscript, we performed additional reciprocal IPs of each XRCC4 family protein expressed at endogenous levels to examine their association with endogenous Pol λ . Our new results confirm that Pol λ associates with XRCC4, XLF and PAXX in cells expressing endogenous levels of each protein (Fig. 3B).

2. Page 18, line 418. "PAXX^{-/-}, 418 XLF^{-/-} and XRCC4^{-/-} U2OS cells", U2OS cells are aneuploidy.

We have corrected our original designation of these U2OS cell lines and subsequently refer to these as either PAXX KO, XLF KO, XRCC4 KO or PAXX/XLF DKO cells throughout the manuscript. For each clonal knockout cell line generated using CRISPR-Cas9, we performed both immunoblotting and DNA sequencing of genomic DNA surrounding guide sequences. Each knockout cell line was completely deficient in the indicated protein (Fig. 5D and 10A). We noted only 2 distinct indels were identified for each clonal gene knockout cell line generated, with at least 6 independent colonies analysed by DNA sequencing (Supplementary Table 4).

3. Fig.3C, it's worth testing whether XLF also have similar activity.

We performed similar EMSA assays by replacement of full length PAXX with XLF in the presence or absence of Ku and Pol λ . To eliminate binding of XLF directly to DNA in a length-dependent manner, we used a 33 bp rather than a 90 bp IR700-labeled dsDNA which previously used in other experiments shown in Fig. 4 (9). Similar to our results using PAXX, binding of XLF to Pol λ using EMSA *in vitro* required Ku (Figs. 4C and D, *left panels*). Furthermore, a C-terminal XLF deletion mutant, in which the last 66 amino acids were deleted including a basic amino acid-rich Ku-binding motif, did not bind Ku (Fig. 4D, *right panel*, compare lanes 3 and 5) and did not supershift Ku-Pol λ complexes. In summary, these new results show that XLF, similar to PAXX, predominantly interacts with Pol λ in a Ku-dependent manner and requires the C-terminal 66 amino acid region of XLF.

4. Fig. 5D, the quality of this figure is low.

We have rescanned data shown in original Fig. 5D (new Fig. 6D) to improve its quality.

5. Fig. 5B, G&F. To excluding that the stimulation activity came from contamination, it's necessary to include a point mutant of PAXX, which does not interact with Pol λ , as a negative control.

Our new results (see above response to Major Point 1) show that the N-terminal head domain of PAXX (and also XLF and XRCC4), similar to the full length protein, stimulates Pol λ -dependent gap filling activity correlating with its ability to interact, albeit weakly, with the Ser-Pro-8kDa region of Pol λ (Figs. 7B, D-E and G, Fig. 8C-D). Notably, the C-terminal portion of PAXX (CC-CTR) inhibits Pol λ -dependent gap filling activity (Fig. 7B). In addition, we observe a similar ability of human XRCC4 family proteins to enhance Pol λ -dependent gap filling activity irrespective of their expression in either human HEK293F or *E.Coli*, suggesting it is unlikely that stimulation results from contamination (Figs. 6B-C, Figs. 7B, D and G, Fig. 8B-C, Supplementary Fig. 6).

6. Fig. 6, it's also better to use a point mutant of PAXX as a negative control.

While we agree with the Reviewer's suggestion, additional ligation assays including a PAXX-V199A/F201A mutant were not performed. In a previously published study, Xing and colleagues showed that PAXX-F201A, a PAXX mutant which did not interact with DNA-bound Ku, did not stimulate XLF-dependent ligation of non-cohesive DNA ends in contrast to PAXX-WT (10)

Reviewer #3 (Remarks to the Author):

In this manuscript Craxton et al. dissect the role and interactions of PAXX in the non-homologous end joining (NHEJ) pathway and identify Pol L as a major downstream factor. First, they generate the PAXX interactome in parallel with its paralogs XRCC4 and XLF, confirming association of PAXX with NEHJ factors, including Pol L, preferentially in soluble chromatin and the overlap with XLF and XCRR4. Next, they focus in the interaction of PAXX and DNA polymerase Pol L. Pol L interacts with PAXX and its paralogs and PAXX interaction is dependent of DNA and Ku70/80. Next, using laser-induced DSBs they find that PAXX or its paralogs are required for the recruitment in vivo of Pol L but not for Ku recruitment and retention. When they assay Pol L gap filling activity in vitro they find that either PAXX or its paralogs enhances Pol L activity, and the absence of PAXX or any of its paralogs reduced Pol L activity when Pol L is immunoprecipitated from cells. Using Pol L mutants lacking functional domains they dissect how PAXX and its paralogs enhance the gap-filling activity of Pol L, and in the case of PAXX this is independent of its interaction with Ku. Neither PAXX nor XLF have an effect on Pol Mu activity. Finally, using a qPCR ligation assay with a substrate with distinct DNA ends they found that joining depends on the presence of Ku70/80, and Pol L and PAXX have cooperative effects in stimulating ligation of non-compatible ends.

The paper provides an advance in our understanding of NHEJ, by revealing a new link between PAXX and gap filling activities. The datasets are generally convincing, but the author should try to address the following points:

What is the role of Ku in the gap filling *in vitro* experiments? Ku is essential for *in vitro* interaction of PAXX with Pol λ , and essential for the joining in the qPCR ligation assay, but in there is no Ku dependency in the gap-filling experiments and the mutant PAXX-VF enhances Pol L in the same way than the WT.

In our original submission, we did not include Ku in the *in vitro* DNA polymerase λ gap filling assays. Indeed, our new studies showed that Ku has a minor effect on Pol λ -dependent gap filling activity (although Ku is essential in end-joining ligation reactions as shown in Fig. 9) with the short DNA substrates (33-mer with a 5 nt gap) used in our *in vitro* assays (Supplementary Fig. 5C). All subsequent experiments for our revised manuscript were performed in the presence of Ku as stated in the additional figures and accompanying legends. To further exclude Ku contribution to stimulation of Pol λ *in vitro* we used a PAXX mutant that does not bind Ku, PAXX-V199A/F201A (PAXX-VF) (Fig. 3B, (6)). Notably PAXX-VF stimulated Pol λ -dependent gap filling activity comparable to PAXX-WT even in the presence of Ku (Supplementary Fig. 6B). These conclusions were further supported by the ability of C-terminal deletion mutants of PAXX or XLF to stimulate Pol λ -dependent gap filling activity comparable to full length PAXX or XLF (Supplementary Fig. 6A, C and D). Importantly, these C-terminal deletions include the basic-rich Ku binding region within either PAXX or XLF.

How is possible that PAXX enhances Pol λ gap filling *in vitro* without this interaction if PAXX is dependent on KU? (in the Methods they don't say if Ku is present in the gap filling experiments)

As stated above Ku does not appear to bear any major effect on the stimulation of the Pol λ enzymatic activity as measured by gap-filling reactions (although Ku is essential in end-joining ligation reactions as shown in Fig. 9). Instead new experiments included in the revised manuscript identify a novel direct interaction between the head domains of XRCC4 family proteins and Pol λ SP/8kD domain (Fig. 6E). Importantly, these head domain regions

strongly enhance Pol λ -dependent gap filling activity via a mechanism independent of their binding to Ku (Fig. 6B and D; Fig. 7C-D). These new data are summarized by a cartoon illustrating the main novel findings in this study (Fig. 10C).

Which interacting proteins are only associated with PAXX and exclusive of XLF (or *vice versa*)? Is there any upstream or downstream factor that might point to a non-redundant role between PAXX and XLF during DNA repair ?

We have included an additional table (Supplemental Table 7) listing interacting proteins which are associated with PAXX (but not XLF) or *vice versa* and added a sentence in the Results section. We state that “further studies are required to identify whether these proteins may contribute to non-redundant roles of either PAXX or XLF” as we currently do not have experimental evidence suggesting that any of these proteins may serve as unique upstream/downstream factors of PAXX or XLF.

With the qPCR ligation assay they demonstrate that PAXX synergize with XLF in the joining of non-compatible ends only in the presence of small gaps or blunt ends. They show the % of joining with and without PAXX in the presence of the rest of components of NHEJ. To better compare the role of PAXX and XLF in the joining it would be interesting to assess the % of joining with and without XLF in the presence of the other components of NHEJ.

We concur with the Reviewer and to emphasise these results have added a sentence to the “Results” section which states “Joining of all combinations of DNA ends tested was strongly enhanced by XLF (Fig. 6A-E, compare lanes 6 and 10), whereas PAXX appeared to increase joining of blunt ends with 3’ overhangs only (Fig. 6A-E, compare lanes 10 and 11).”

Do they have data on the Pol λ interactome?

We performed additional mass spectrometry studies using FLAG-Pol λ as a bait protein. These experiments were performed using the same experimental conditions as our data shown in Figs. 1-2, Supplementary Tables 5, 6 and 9, except that FLAG-Pol λ was transiently expressed. Our results show that FLAG-tagged Pol λ interacts with multiple NHEJ factors including XRCC4 family proteins (Supplementary Tables 8 and 10). These results are

therefore consistent with our IP experiments showing that XRCC4 family proteins associate with endogenous Pol λ (Figs. 3A and B). Furthermore, these experiments complement our findings that reciprocal studies using FLAG-tagged PAXX as bait combined with detection by mass spectrometry or immunoblotting also showed that Pol λ interacts with PAXX (Supplementary Fig. 2A and B). We have added a new Table (Supplementary Table 8), which shows the NHEJ-related proteins identified by mass spectrometry and also include complete details of all proteins identified by mass spectrometry (Supplementary Table 10).

Minor:

- Make 1C-D more clear in the labels, it is difficult to read. Highlight PP6, DYN1-3, TRF2, RAP1.

To highlight these PAXX-, XLF-, XRCC4- and DNA-PKcs-interacting proteins which we subsequently refer to in the Results section (lines 334-342) we have added shaded boxes (light yellow) in the supplementary Excel files which show proteins identified by mass spectrometry. In addition, we have significantly enlarged the original cluster diagrams (original Figs. 1C and D) which are now shown as full size Figs. 1 and 2.

- Cropped supplementary 2 labels.

We have carefully checked Supplementary Fig. 2 for cropped labels and changed accordingly. In addition, we have enlarged font sizes of some labels to enhance legibility e.g. Supplementary Fig. 2A and 2D.

- In supplementary figure 3 C, why does the FLAG IP with PAXX-WT not immunoprecipitate any NHEJ component and it is the same as the mutant VF?

We assume the Reviewer is referring to supplementary Fig. 3D, which shows FLAG IPs of PAXX-WT and a PAXX-VF mutant, which does not bind DNA-bound Ku70/80 heterodimers. Neither of these proteins associate with other NHEJ proteins as the immunoprecipitations were performed in the presence of high NaCl concentrations (0.6M) and a short time of M2 bead capture to specifically isolate highly purified, preferably homogenous, preparations of

FLAG-PAXX-WT and a –VF mutant for *in vitro* studies (EMSA, DNA polymerase enzymatic assays, ligation assays). We have added text to clarify this point, stating “To assess a role for PAXX-Ku interaction(s) in formation of the PAXX-Pol λ -Ku70/80-DNA quaternary complex, we generated and purified to homogeneity a PAXX mutant in which two highly conserved C-terminal residues were mutated to alanine as reported (PAXX-V199A/F201A (PAXX-VF)) Supplementary Fig. 3D (6)).”

1. Liu, X., Shao, Z., Jiang, W., Lee, B. J., and Zha, S. (2017) PAXX promotes KU accumulation at DNA breaks and is essential for end-joining in XLF-deficient mice. *Nature communications* **8**, 13816
2. Blackford, A. N., and Jackson, S. P. (2017) ATM, ATR, and DNA-PK: The Trinity at the Heart of the DNA Damage Response. *Molecular cell* **66**, 801-817
3. Pryor, J. M., Waters, C. A., Aza, A., Asagoshi, K., Strom, C., Mieczkowski, P. A., Blanco, L., and Ramsden, D. A. (2015) Essential role for polymerase specialization in cellular nonhomologous end joining. *Proceedings of the National Academy of Sciences of the United States of America* **112**, E4537-4545
4. Tsai, C. J., Kim, S. A., and Chu, G. (2007) Cernunnos/XLF promotes the ligation of mismatched and noncohesive DNA ends. *Proceedings of the National Academy of Sciences of the United States of America* **104**, 7851-7856
5. Chang, H. H., Watanabe, G., Gerodimos, C. A., Ochi, T., Blundell, T. L., Jackson, S. P., and Lieber, M. R. (2016) Different DNA End Configurations Dictate Which NHEJ Components Are Most Important for Joining Efficiency. *The Journal of biological chemistry* **291**, 24377-24389
6. Ochi, T., Blackford, A. N., Coates, J., Jhujh, S., Mehmood, S., Tamura, N., Travers, J., Wu, Q., Draviam, V. M., Robinson, C. V., Blundell, T. L., and Jackson, S. P. (2015) DNA repair. PAXX, a paralog of XRCC4 and XLF, interacts with Ku to promote DNA double-strand break repair. *Science* **347**, 185-188
7. Tadi, S. K., Tellier-Lebegue, C., Nemoz, C., Drevet, P., Audebert, S., Roy, S., Meek, K., Charbonnier, J. B., and Modesti, M. (2016) PAXX Is an Accessory c-NHEJ Factor that Associates with Ku70 and Has Overlapping Functions with XLF. *Cell reports* **17**, 541-555
8. Sato, Y., Kameya, M., Arai, H., Ishii, M., and Igarashi, Y. (2011) Detecting weak protein-protein interactions by modified far-western blotting. *Journal of bioscience and bioengineering* **112**, 304-307
9. Lu, H., Pannicke, U., Schwarz, K., and Lieber, M. R. (2007) Length-dependent binding of human XLF to DNA and stimulation of XRCC4.DNA ligase IV activity. *The Journal of biological chemistry* **282**, 11155-11162
10. Xing, M., Yang, M., Huo, W., Feng, F., Wei, L., Jiang, W., Ning, S., Yan, Z., Li, W., Wang, Q., Hou, M., Dong, C., Guo, R., Gao, G., Ji, J., Zha, S., Lan, L., Liang, H., and Xu, D. (2015) Interactome analysis identifies a new paralogue of XRCC4 in non-homologous end joining DNA repair pathway. *Nature communications* **6**, 6233

REVIEWERS' COMMENTS:

Reviewer #1 (Remarks to the Author):

The authors have made significant revisions, but important flaws remain.

One of the central concerns of two reviews was a possible undue emphasis on PAXX. The revised manuscript addresses this, and shows either XLF or PAXX (or even XRCC4, in some contexts) can contribute to Pol lambda activity, although there is not always perfect concordance when comparing paralogs. Unfortunately, this undercuts their argument for significance. It has long been appreciated that XLF is required for significant Pol lambda activity. Since PAXX is redundant to XLF for many functions, it is probably not surprising that stimulation of Pol lambda activity can be added to this list.

Radiosensitivity data added in the revision in an effort to address significance issues is also problematic. This data was not analyzed statistically. Consider using e.g. methods described *Radiat. Oncol.*, 10:223 (2015). It thus isn't clear if there is any significant difference in radiosensitivity in the absence of Pol lambda, regardless of context. What differences are observed are at best mild.

There is still no direct evidence that PAXX interaction significantly influences Pol lambda function during NHEJ in cells. There is no evidence that PAXX interaction is required for change in NHEJ products like that shown in Pol lambda deficient cells in e.g. *PLoS ONE*. 6, e28756 (2011). It is therefore not clear the differences observed in in vitro studies will be observed during cellular repair.

Regarding the statement (top of page 20) – “Loss of PAXX or depletion of either XLF or XRCC4 resulted in reduced gap filling activity in Pol I IPs, demonstrating a synergy between PAXX paralogs in promoting Pol I activity in cells (Fig. 6D).” This is misleading, since the experiment employs cell extracts, and not live cells. The result is also overstated, since there is no evidence the slight reductions observed after factor depletion is reproducible.

The authors sometimes detect more joining when they add PAXX and XLF together than when adding one or the other alone, and refer to this as synergy. They never adequately resolved whether this was synergy, or simply a cumulative effect of adding twice the amount of an XRCC4 paralog, where the identity of the paralog is irrelevant. For example, adding twice as much PAXX alone (or XLF alone) may be just as effective as adding the mixture. Supplementary Figure 8 does not adequately address this issue.

Figure 1 is now legible. Unfortunately, Figure 2 still cannot be read, regardless of which source of figure was viewed, because of low contrast (black on red) text. Higher contrast should be considered.

Reviewer #2 (Remarks to the Author):

In general, the authors have taken my critiques into a fine revision. I support the manuscript's acceptance.

Reviewer #3 (Remarks to the Author):

The authors have adequately addressed the reviewers comments. The results are novel,

interesting and make an solid contribution to the DNA repair field. I recommend publication in Nature Communications.

REVIEWERS' COMMENTS:

Reviewer #1 (Remarks to the Author):

The authors have made significant revisions, but important flaws remain.

One of the central concerns of two reviews was a possible undue emphasis on PAXX. The revised manuscript addresses this, and shows either XLF or PAXX (or even XRCC4, in some contexts) can contribute to Pol lambda activity, although there is not always perfect concordance when comparing paralogs. Unfortunately, this undercuts their argument for significance. It has long been appreciated that XLF is required for significant Pol lambda activity. Since PAXX is redundant to XLF for many functions, it is probably not surprising that stimulation of Pol lambda activity can be added to this list.

We concur with the Reviewer that PAXX and XLF appear to share many functions based upon genetic studies published to date. Our manuscript now clearly defines one specific shared function of PAXX and XLF and the relevant molecular mechanisms involved. Accordingly, in this work, we show that all XRCC4 paralogs can stimulate Pol λ activity via weak interaction with their head domains, thereby revealing a new common molecular mechanism by which XRCC4 family proteins promote the gap-filling activity of Pol λ .

Radiosensitivity data added in the revision in an effort to address significance issues is also problematic. This data was not analyzed statistically. Consider using e.g. methods described Radiat. Oncol., 10:223 (2015). It thus isn't clear if there is any significant difference in radiosensitivity in the absence of Pol lambda, regardless of context. What differences are observed are at best mild.

As suggested by the Reviewer we statistically analysed radiosensitivity data shown in Fig. 10B using a two-tailed paired t-test. All p-values have now been included in the legend accompanying Fig. 10B. These results show that depletion of Pol λ significantly increased cellular radiosensitivity, albeit we agree with the reviewer that the differences are relatively mild consistent with the fact that Pol λ activity can be masked in cells by Pol μ .

There is still no direct evidence that PAXX interaction significantly influences Pol lambda function during NHEJ in cells. There is no evidence that PAXX interaction is required for change in NHEJ products like that shown in Pol lambda deficient cells in e.g. PLoS ONE. 6, e28756 (2011). It is therefore not clear the differences observed in in vitro studies will be observed during cellular repair.

We would like to highlight that our study shows defective recruitment and retention of Pol λ at laser-induced DSB sites *in vivo* in PAXX-, XLF- and XRCC4 KO cells (Fig. 5F). These results imply a non-redundant requirement for each XRCC4 family protein in Pol λ function *in vivo*, although we acknowledge various mechanisms may contribute to this finding including defects in XLF/XRCC4 filaments and interaction(s) with other NHEJ factors such as Ku. We acknowledge further

experiments are required in order to establish the relative contributions of direct and indirect (via binding to Ku) interaction of PAXX (and indeed XLF and XRCC4) with Pol λ to NHEJ *in vivo*. Accordingly, we have included this sentence in the Discussion emphasising these concerns.

Regarding the statement (top of page 20) – “Loss of PAXX or depletion of either XLF or XRCC4 resulted in reduced gap filling activity in Pol I IPs, demonstrating a synergy between PAXX paralogs in promoting Pol I activity in cells (Fig. 6D).” This is misleading, since the experiment employs cell extracts, and not live cells. The result is also overstated, since there is no evidence the slight reductions observed after factor depletion is reproducible.

In accordance with Reviewer 1, we have changed the text from “cells” to “derived from cell extracts *in vitro*” (page 21, line 1 (originally top of page 20). We have also changed the concluding sentence of this paragraph and now state that “Loss of PAXX or depletion of either XLF or XRCC4 resulted in reduced gap filling activity in Pol λ IPs, demonstrating a role for PAXX paralogs in promoting Pol λ activity (Fig. 6D).”

The authors sometimes detect more joining when they add PAXX and XLF together than when adding one or the other alone, and refer to this as synergy. They never adequately resolved whether this was synergy, or simply a cumulative effect of adding twice the amount of an XRCC4 paralog, where the identity of the paralog is irrelevant. For example, adding twice as much PAXX alone (or XLF alone) may be just as effective as adding the mixture. Supplementary Figure 8 does not adequately address this issue.

We realised that we use the word "synergy" in the manuscript in two different contexts and meanings. In the title of the paper we state: " PAXX and its paralogues **synergistically** direct... " to indicate that these factors act in concert and at various levels, as described in the manuscript, to facilitate the activity of Pol λ . In the context of the data presented in Figure 9 we have used the term synergy to describe a potentially non-linear (synergistic) effect of combining two PAXX paralogues in the reaction. This is the context where Reviewer 1 raises his objection. We acknowledge this specific point by Reviewer 1 and have replaced any reference to "synergy" (e.g. “PAXX and XLF together synergised with Pol λ to promote joining of blunt ends with 2-4bp 3' overhangs” in the manuscript text, Figure 9 title, abstract and discussion) by more appropriate terms ("enhanced joining activity between PAXX, XLF and Pol λ " or " PAXX, XLF and Pol λ co-operated").

Figure 1 is now legible. Unfortunately, Figure 2 still cannot be read, regardless of which source of figure was viewed, because of low contrast (black on red) text. Higher contrast should be considered.

We have applied the same colour scheme used in Fig. 1 to Fig. 2 (black text with yellow shaded boxes indicating interacting proteins with a specific bait protein (subtitled PAXX, XLF, XRCC4, PRKDC) and light grey boxes showing proteins which did not interact with the specific bait protein but

interacted with one or more of the other 3 bait proteins. In our opinion and as indicated for new Figure 1 by Reviewer 1, this colour scheme significantly improves the legibility.

Reviewer #2 (Remarks to the Author):

In general, the authors have taken my critiques into a fine revision. I support the manuscript's acceptance.

Reviewer #3 (Remarks to the Author):

The authors have adequately addressed the reviewers comments. The results are novel, interesting and make a solid contribution to the DNA repair field. I recommend publication in Nature Communications.